# ANGLE K-MEANS

**Shenfei Pei[1], Ruiyu Huang[1], Yiqing Hu[1], Zhongqi Lin[1], Xudong Jiang[2], Zengwei Zheng[1]\***
[1]Hangzhou City University, China
[2]Nanyang Technological University, Singapore
`shenfeipei@gmail.com, 32309235@stu.hzcu.edu.cn`
`32303023@stu.hzcu.edu.cn, linzq@hzcu.edu.cn`
`EXDJiang@ntu.edu.sg, zhengzw@hzcu.edu.cn`

## ABSTRACT

We propose an accelerated exact $k$-means algorithm, *Angle k-means*. As its name suggests, the algorithm mainly leverages angular relationships between data points and cluster centers to reduce computational overhead. Although grounded in straightforward geometric principles, it delivers substantial performance improvements in empirical evaluations. In contrast to existing acceleration techniques, our model introduces no new hyperparameters, preserving full compatibility with standard $k$-means. Theoretical analysis shows that *Angle k-means* maintains linear time complexity with respect to both sample size and dimensionality, while empirical evaluations on diverse real-world datasets demonstrate significant speedup over state-of-the-art algorithms such as ball $k$-means and Exp-ns.

## 1 INTRODUCTION

Clustering is a fundamental task in machine learning and continues to receive increasing attention. Due to its unsupervised nature, clustering algorithms are frequently employed to process large-scale and unlabeled data samples which can be easily collected from the Internet.

For example, in industrial IoT networks, clustering are used to improve the energy management of wireless sensors, thereby extending network lifespan and reducing energy consumption (Saeedi et al., 2024). Besides this, various clustering algorithms have been proposed recently and applied in many fields, such as document classification (Dodda & Babu, 2024; Diallo et al., 2022; Liu et al., 2025), power systems (Miraftabzadeh et al., 2023), urban global positioning system (Ran et al., 2024), and emotion categorization (Zhou et al., 2023; Salloum et al., 2024).

Although numerous clustering algorithms have emerged in recent years, Lloyd's $k$-means algorithm (Lloyd, 1982) remains extensively utilized due to its computational efficiency and conceptual simplicity. The algorithm iterates through three fundamental steps:

- **Initialization:** Select initial cluster centers
- **Assignment:** Assign each sample to its nearest cluster center
- **Update:** Recompute cluster centroids based on current assignments
- Iterate "Assignment" and "Update" steps until convergence

Two primary limitations affect standard implementations: (1) Performance exhibits high sensitivity to random center initialization, and (2) The assignment step requires calculating *all* sample-center distances, becoming computationally prohibitive for large datasets. This work specifically targets acceleration of the assignment step by reducing required distance computations.

Our principal contributions are:

- We propose *Angle k-means*, an accelerated exact $k$-means variant. Through precomputation of geometric auxiliary variables (including the distance and angles between cluster centers), the center closest to the sample can be found without calculating the distance between the sample and all cluster centers.

---

\*Corresponding author.

- No hyper-parameters are introduced in our model and it enjoys the same usage as the standard $k$-means, which makes our algorithm more user-friendly.

- Comprehensive evaluation across 14 real-world datasets demonstrates significant speedups over state-of-the-art alternatives (including Ball $k$-means and Exp-ns) while maintaining identical clustering accuracy.

## 2 RELATED WORK

This section reviews accelerated *exact* $k$-means algorithms, focusing primarily on tree-based and boundary-based acceleration techniques. We also briefly discuss relevant approximate methods.

**Tree-based acceleration:** Kanungo et al. (Kanungo et al., 2002) introduced a filtering algorithm where samples are stored in a static kd-tree. Samples within the same tree node share candidate centers, avoiding tree reconstruction during center updates. Building on cover tree foundations, Yang et al. (Lang & Schubert, 2023) demonstrate that ball covers in cover trees produce tighter bounds than the bounding boxes used in k-d tree approaches. Their proposed *Cover means* algorithm achieves robust performance across wider parameter ranges compared to traditional tree-based methods.

For scenarios with large cluster counts, Curtin et al. (Curtin, 2017) proposed dual-tree k-means where both data and centers are stored in cover trees. Under certain assumptions, this achieves $O(n + k \log k)$ per-iteration complexity and is agnostic to tree structure. Despite these advances, tree-based methods exhibit degraded performance in high-dimensional spaces due to the curse of dimensionality. See (Aljabbouli et al., 2020; Wang et al., 2020) for related algorithms.

**Bound-based methods:** Elkan's algorithm (Elkan, 2003) uses triangle inequality to maintain $k$ lower bounds per sample, incurring $O(nk)$ memory overhead that scales poorly with cluster count. Ding et al (Ding et al., 2015) proposed the Yinyang $k$-means algorithm, which not only clusters samples but also centers. When clustering the centers, the standard $k$-means algorithm is used, and the number of clusters is a hyper-parameter. The author suggests that its value should not be greater than $k/10$. It is worth noting that this method not only accelerates the step of "Assignment" but also accelerates the step of "Update".

Hamerly (Hamerly, 2010) presented a modification and simplification version of Elkan's $k$-means algorithm. It does not store the lower bounds for all sample-center combinations, so it does not require $O(nk)$ memory. In fact, the total space complexity of this algorithm is $O(n)$. Drake et al. (Drake & Hamerly, 2012) combined Hamerly's algorithm with Elkan's algorithm and proposed Adaptive $k$-means. In Elkan's and Hamerly's algorithms, each sample has $k$ and 1 lower bound, respectively, while in adaptive $k$-means, each sample has $b$ ($1 < b < k$) lower bounds, where $b$ is a hyperparameter, that's why it's called adaptive $k$-means. Annulus $k$-means (Drake, 2013) can be regarded as a variant of Hamerly's algorithm. Given a sample $\mathbf{x}$, Annulus $k$-means finds the candidate centers by $|\|\mathbf{x}\|_2 - \|\mathbf{c}\|_2| \leq \|\mathbf{x} - \mathbf{c}\|_2$, where $\mathbf{c}$ is the nearest center previously assigned to $\mathbf{x}$, and then finds the center closest to $\mathbf{x}$ among these candidate centers. Because the region where the candidate centers are located is a ring, it is called Annulus $k$-means. Exponion algorithm (Newling & Fleuret, 2016) is also a variant of Hamerly's algorithm. Unlike Annulus $k$-means, Exponion looks for candidate centers in a sphere with center $\mathbf{c}$ instead of the coordinate origin for sample $\mathbf{x}$, which results in $k^2$ distances to be considered by Exponion. In addition, in (Newling & Fleuret, 2016), the authors also proposed a technique to make the bounds tighter, which can be applied to all boundary-based algorithms. Based on this technique, the author proposed an improved version of Exponion, known as Exp-ns. Xia et al. (Xia et al., 2022) proposed Ball $k$-means and provided both ring and no-ring versions. It characterizes each cluster with a ball and defined the concept of cluster neighbors. For each sample, it finds the cluster center closest to the sample among the cluster neighbors. In addition, the author also defined a stable region for each cluster, and assignments of samples in the stable region will not be changed. Ismkhan et al. (Ismkhan & Izadi, 2022) proposed KM-G*, an accelerated k-means variant that utilizes geometric lower bounds to reduce distance computations. For more related algorithms, please refer to (Cohen-Addad et al., 2020; Hirschberger et al., 2021; Zhang et al., 2024).

**Approximate $k$-means:** Deng et al. (Deng & Zhao, 2018) proposed a fast $k$-means variant using a $k$-nearest neighbor graph, where the set of candidate centers is composed of the nearest centers assigned to $\mathbf{x}$'s neighbors previously, for sample $\mathbf{x}$. Usually, the number of nearest neighbors is

smaller than the number of clusters, so it has low computational complexity. A fast heuristic method is proposed (Khandelwal & Awekar, 2017), where only a subset of the nearby clusters is considered as candidates to update the labels of the sample. Grandoni et al. (Grandoni et al., 2022) developed a refined approximation method for the Euclidean k-means, which contributes to more precise clustering results. More algorithms can be found here (Wen et al., 2022; Du et al., 2024). On improving the quality of clustering, the Self-Constrained Clustering Ensemble (Wei et al., 2025) integrates self-supervision and metric learning to iteratively refine base clusterings and improve ensemble quality.

## 3 OUR MODEL

In this section, we first review the k-means problem, and then we introduce a optimization algorithm to solve it, called Angle $k$-means (AKM). Our algorithm can be seen as a fast version of Lloyd's algorithm, so Lloyd's algorithm will also be briefly reviewed. To introduce AKM, some definitions and lemmas are given first, then a special case of AKM is given to introduce the model motivation.

Let $\mathbf{X} = [\mathbf{x}_1, \cdots, \mathbf{x}_n]^T \in \mathbb{R}^{n \times d}$ denote the dataset to be clustered, where $n$ and $d$ denote the number of samples and features respectively, $\mathbf{y} \in \mathbb{R}^n$ the labels of the samples (i.e, $y_i = g$ means that $\mathbf{x}_i$ belongs to the $g$-th cluster), $\mathbf{C} = [\mathbf{c}_1, \cdots, \mathbf{c}_k]^T \in \mathbb{R}^{k \times d}$ the centers of clusters, where $k$ denotes the number of clusters, which is a positive integer known in advance, the $k$-means problem can be formulated as follows:

$$\min_{\mathbf{y}} \sum_{g=1}^{k} \sum_{i \in \mathcal{A}_g} \|\mathbf{x}_i - \mathbf{c}_g\|_2^2 \tag{1}$$

where $\mathcal{A}_g = \{j \mid y_j = g\}, \mathbf{c}_g = \frac{1}{|\mathcal{A}_g|} \sum_{i \in \mathcal{A}_g} \mathbf{x}_i$

To solve problem (1), many efficient algorithms have been proposed, among which the Lloyd's algorithm proposed in Lloyd (1982) is widely used. The main steps involved can be expressed as: 1. Update labels $\mathbf{y}$ according to the centers $\mathbf{C}$. 2. Calculate the centers $\mathbf{C}$ according to $\mathbf{y}$. That is

$$y_i = \operatorname*{arg\,min}_{j \in \{1, \cdots, k\}} \|\mathbf{x}_i - \mathbf{c}_j\|_2^2, \quad \forall i = 1, \cdots, n. \tag{2}$$

$$\mathbf{c}_g = \frac{1}{|\mathcal{A}_g|} \sum_{i \in \mathcal{A}_g} \mathbf{x}_i, \quad \forall g = 1, \cdots, k. \tag{3}$$

In this paper, we mainly focus on the first step of Lloyd's algorithm. Specifically, by pre-calculating some auxiliary variables, such as the distance and angles between centers, etc., the center closest to sample $\mathbf{x}_i$ can be found without calculating the distance between $\mathbf{x}_i$ and all centers.

### 3.1 ANGLE $k$-MEANS

For convenience, some lemmas and notations are first given before introducing the AKM algorithm.

**Lemma 1.** $\mathbf{p}$ and $\mathbf{q}$ are two points in the Euclidean space $\mathbb{R}^n$, $r_1$ and $r_2$ are constants greater than 0, then the optimal value of the following problem is $|r_1 - r_2|$.

$$\min_{\mathbf{p}, \mathbf{q}} \quad \|\mathbf{p} - \mathbf{q}\|_2$$
$$s.t. \quad \|\mathbf{p}\|_2 = r_1, \|\mathbf{q}\|_2 = r_2 \tag{4}$$

*Proof.* See supplementary material. □

**Theorem 2.** $\mathbf{a}$ and $\mathbf{b}$ are two points in the Euclidean space $\mathbb{R}^n$, $\mathbf{a} \neq \mathbf{b}$, $e_1, e_2, e_3, e_4$ are four constants greater than 0. Let $d(\mathbf{p}, \mathbf{q})$ denotes the optimal value of the following problem

$$\min_{\mathbf{p}, \mathbf{q}} \quad \|\mathbf{p} - \mathbf{q}\|_2$$
$$s.t. \begin{cases} \|\mathbf{p} - \mathbf{a}\| = e_1, \|\mathbf{p} - \mathbf{b}\| = e_2 \\ \|\mathbf{q} - \mathbf{a}\| = e_3, \|\mathbf{q} - \mathbf{b}\| = e_4 \end{cases} \tag{5}$$

*we have*

$$d(\mathbf{p}, \mathbf{q}) = \sqrt{e_2^2 + e_4^2 - 2e_2 e_4 \cos(\theta_1 - \theta_2)} \tag{6}$$

*where $\theta_1 = \angle \mathbf{abp}$, $\theta_2 = \angle \mathbf{abq}$.*

*Proof.* Since the operations of rotation and translation are both distance-preserving transformations, we assume that $\mathbf{b} = \mathbf{0}$, $\mathbf{a} = (-r, 0, \cdots, 0)$, where $r = \|\mathbf{a} - \mathbf{b}\|$ is a constant. So, we have:

$$\begin{cases} p_1 = -e_2 \cos\theta_1 \\ p_2^2 + \cdots + p_n^2 = e_2^2 \sin^2\theta_1 \end{cases} \text{ and } \begin{cases} q_1 = -e_4 \cos\theta_2 \\ q_2^2 + \cdots + q_n^2 = e_4^2 \sin^2\theta_2 \end{cases}$$

Let $\hat{\mathbf{p}} = (p_2, \cdots, p_n)$, $\hat{\mathbf{q}} = (q_2, \cdots, q_n)$, we have

$$\|\mathbf{p} - \mathbf{q}\|_2^2 = (p_1 - q_1)^2 + \|\hat{\mathbf{p}} - \hat{\mathbf{q}}\|_2^2 \tag{7}$$

According to Lemma 1, it's easy to find the minimum Euclidean distance between $\mathbf{p}$ and $\mathbf{q}$. $\qquad \square$

In order to show our algorithm more clearly, some notations used in this section are introduced first.
1. $\mathbf{r} = [r_1, \cdots, r_n]^T \in \mathbb{R}^n$, where $r_i > 0$ represents the Euclidean distance between $\mathbf{x}_i$ and the center of the cluster to which it belongs. i.e., $r_i = \|\mathbf{x}_i - \mathbf{c}_g\|_2$, where $g = y_i$.
2. $\boldsymbol{\beta} = [\beta_1, \cdots, \beta_n]^T \in \mathbb{R}^n$, where $\beta_i$ is the angle formed by $\overrightarrow{\mathbf{oc}_g}$ and $\overrightarrow{\mathbf{c}_g\mathbf{x}_i}$, i.e., $\beta_i = \angle\mathbf{oc}_g\mathbf{x}_i$. 3. $\boldsymbol{\Theta} = [\boldsymbol{\theta}_1, \cdots, \boldsymbol{\theta}_k]^T \in \mathbb{R}^{k \times k}$, where $\theta_{ij}$ denotes the angle formed by line $\overrightarrow{\mathbf{oc}_i}$ and $\overrightarrow{\mathbf{c}_i\mathbf{c}_j}$.

The core idea of this paper is to reduce the number of candidate centers, for each sample. Specifically, some centers that are far away from the sample are excluded by some computations with lower complexity.

Given $\mathbf{x}_i$, denote $y_i$ as $g$, combined with the above symbols, we consider the following two questions:

- **Case 1**: If $\|\mathbf{c}_g - \mathbf{c}_j\| \geq r_i$, and $|\beta_i - \theta_{gj}| \geq \pi/3$. Does $\|\mathbf{x}_i - \mathbf{c}_j\| \geq r_i$ holds?
- **Case 2**: If $\|\mathbf{c}_g - \mathbf{c}_j\| \leq r_i$, and $|\beta_i - \theta_{gj}| \geq \pi/2$. Does $\|\mathbf{x}_i - \mathbf{c}_j\| \geq r_i$ holds?

The answer to both questions is yes. Next, we give a proof for Case 1. The proof of Problem 2 is omitted here because it is very similar to that of Problem 1.

From Theorem 2, we know that the square of the smallest Euclidean distance between $\mathbf{x}_i$ and $\mathbf{c}_j$ is

$$\|\mathbf{x}_i - \mathbf{c}_j\|_2^2 = r_i^2 + \|\mathbf{c}_g - \mathbf{c}_j\|_2^2 - 2r_i\|\mathbf{c}_g - \mathbf{c}_j\|_2 \cos(\beta_i - \theta_{gj}) \tag{8}$$

Note that $\beta_i - \theta_{gj}$ is not $\angle\mathbf{x}_i\mathbf{c}_g\mathbf{c}_j$. This equation is derived from Theorem 2 rather than from the law of cosines.
Since $\cos(\beta_i - \theta_{gj}) \leq 1/2$, we have

$$\|\mathbf{x}_i - \mathbf{c}_j\|_2^2 \geq r_i^2 + \|\mathbf{c}_g - \mathbf{c}_j\|_2^2 - r_i\|\mathbf{c}_g - \mathbf{c}_j\|_2 \tag{9}$$

Since $\|\mathbf{c}_g - \mathbf{c}_j\|_2^2 \geq r_i$, we have

$$\|\mathbf{x}_i - \mathbf{c}_j\|_2^2 \geq r_i^2 + \|\mathbf{c}_g - \mathbf{c}_j\|_2(\|\mathbf{c}_g - \mathbf{c}_j\|_2 - r_i) \geq r_i^2 \tag{10}$$

A schematic of AKM is appears in Figure 1(b). **The points located in the cyan region (case 1) and the purple region (case 2) can be safely ignored when updating the label of $\mathbf{x}_i$ since their distance to $\mathbf{x}_i$ exceeds $r_i$.**

**Corollary 2.1.** *For any $\|\mathbf{c}_g - \mathbf{c}_j\|_2 \geq 0$, there exists $t = \frac{\|\mathbf{c}_g - \mathbf{c}_j\|_2}{2r_i}$ such that for $\cos(\beta_i - \theta_{gj}) \leq t$, we have $\|\mathbf{x}_i - \mathbf{c}_j\|_2 \geq r_i$.*

According to Corollary 2.1, we can judge that some distances between centers and samples are not necessary. Next, we summarize the main steps of Angle $k$-means algorithm. See Algorithm 1 for details.

- **Initialization:** Calculate the distances between the samples and the coordinate origin $\mathbf{o}$, $\mathbf{u} \in \mathbb{R}^n$.
- **Update centers:** Calculate the cluster centers $\mathbf{M}$ according to Eq. (3), as the same as that in Lloyd's.

- **Update angles and distances:** Calculate the distances matrix $\mathbf{D}$, $d_{ij} = \|\mathbf{c}_i - \mathbf{c}_j\|_2$, the distance between the centers and $\mathbf{o}$, $\mathbf{v}$, the angles $\boldsymbol{\beta} \in \mathbb{R}^n$ and $\boldsymbol{\Theta} \in \mathbb{R}^{k \times k}$, according to the cluster centers.

- **Find the set of candidate centers:** For sample $\mathbf{x}_i$, $g = y_i$, we have

$$\mathcal{B}_i = \{l \mid \cos(\beta_i - \theta_{gl}) > d_{gl}/(2r_i), l = 1, \cdots, k\} \tag{11}$$

As shown in Figure 1(c), the distances between the sample $\mathbf{x}_i$ and the centers in the blue area do not need to be calculated.

- **Assignment:** For each sample $\mathbf{x}_i$, find the nearest center in $\mathcal{B}_i$ and then update its label.

Note that: When finding candidate centers for $\mathbf{x}_i$, we visit these centers in order of their distance from $\mathbf{c}_g$, so that we don't have to visit all the centers.

**Time complexity:** From Algorithm 1, it is not difficult to know that: It takes $O(nD)$ time to compute the vector $\mathbf{u}$. Computing the matrix $\mathbf{C}$, $\mathbf{D}$, $\boldsymbol{\Theta}$, vector $\mathbf{v}$, $\mathbf{r}$, and $\boldsymbol{\beta}$ takes $O(nD)$, $O(k^2D)$, $O(k^2)$, $O(kD)$, $O(nD)$, and $O(n)$ time, respectively. Sorting the matrix $\mathbf{D}$ takes $O(k^2D \log k)$ time. $O(npD)$ time is required to update $\mathbf{y}$, where $p$ is the mean of the number of candidate centers for each sample, and $D$ is the number of features. We directly denote the complexity of the loop in Line 15 of Algorithm 1 by $O(nk)$. Therefore, the total time complexity of per-iteration of the proposed method is $O(k^2D \log k + npD + nk)$, where $n$, $k$ are the number of samples and clusters, respectively.

**Space complexity:** In our proposed algorithm, memory is mainly consumed by data matrix $\mathbf{X}$, and variables $\mathbf{C}$, $\mathbf{D}$, and $\boldsymbol{\Theta}$. So the space complexity is $O(k^2)$.

We summarize the time and space complexity of the algorithms involved in the experiments, as shown in Table 1. As can be seen, our algorithm is comparable to these state-of-the-art methods.

## 4 EXPERIMENTS

In this section, we build experiments on several real-world datasets to verify the performance of the proposed algorithm (the number of distances computed and running time). The rest is organized as follows: The datasets used in the experiment are first given, then we introduce the comparison algorithms, and finally, the experimental results are shown.

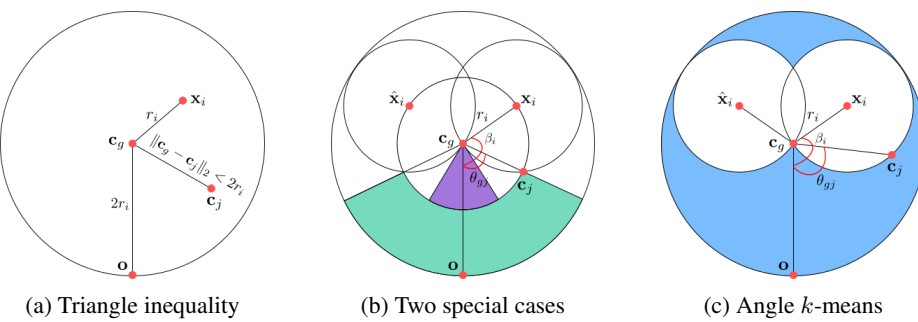

(a) Triangle inequality      (b) Two special cases      (c) Angle $k$-means

Figure 1: $\mathbf{x}_i$ represents the sample that is ready to update its assignment, $\mathbf{c}_g$ denotes the center previously assigned to $\mathbf{x}_i$, $\mathbf{o}$ represents the origin of coordinates. (a): According to the triangle inequality, if $\|\mathbf{x}_i - \mathbf{c}_j\|_2 < r_i$, then $\|\mathbf{c}_g - \mathbf{c}_j\|_2$ must be less than $2r_i$, so we only need to consider the center whose distance from $\mathbf{c}_g$ is less than $2r_i$. (b): Two special cases of Angle $k$-means: If the centers are located in the cyan or purple areas, then the distance from them to $\mathbf{x}_i$ must be greater than or equal to $r_i$. (c): Angle $k$-means: if $\cos(\beta_i - \theta_{gj}) \le \|\mathbf{c}_g - \mathbf{c}_j\|_2/(2r_i)$, i.e., $\mathbf{c}_j$ is located in the blue area, then we have $\|\mathbf{x}_i - \mathbf{c}_j\|_2 \ge r_i$. So there is no need to calculate the distance between $\mathbf{x}_i$ and $\mathbf{c}_j$.

---

**Algorithm 1:** An accelerated exact $k$-means, Angle $k$-means

---

**Data:** $\mathbf{X} \in \mathbb{R}^{n \times d}$, the number of clusters, $c$
**Result:** The labels of samples, $\mathbf{y}$

1 Initialize the labels $\mathbf{y}$ in a random way;
2 Compute $\mathbf{u} \in \mathbb{R}^n$, $u_i = \|\mathbf{x}_i - \mathbf{o}\|_2$, where $\mathbf{o} = \mathbf{0}$ is the coordinate origin;
3 **while** *not converge* **do**
4 $\quad$ Compute the cluster centers, $\mathbf{C} \in \mathbb{R}^{k \times d}$, by Eq. (3);
5 $\quad$ Compute $\mathbf{v} \in \mathbb{R}^k$, $v_i = \|\mathbf{c}_i - \mathbf{o}\|_2$ ;$\mathbf{D} \in \mathbb{R}^{k \times k}$, $d_{ij} = \|\mathbf{c}_i - \mathbf{c}_j\|_2$ ;
6 $\quad$ Obtain the index matrix $\mathbf{H}$ that would sort $\mathbf{D}$, $h_{ij} = g$ means $\mathbf{c}_g$ is the $j$-th neighbor of $\mathbf{c}_i$;
7 $\quad$ Compute $\mathbf{\Theta} \in \mathbb{R}^{k \times k}$ based on $\mathbf{v}$ and $\mathbf{D}$ using the law of cosines ;
8 $\quad$ Compute $\mathbf{r} \in \mathbb{R}^n$, $r_i = \|\mathbf{x}_i - \mathbf{c}_g\|_2$, where $g = y_i$ ;
9 $\quad$ Compute $\boldsymbol{\beta} \in \mathbb{R}^n$ based on $\mathbf{u}$, $\mathbf{v}$ and $\mathbf{r}$ using the law of cosines ;
10 $\quad$ **for** $i = 1, \cdots, n$ **do**
11 $\quad\quad$ $g = y_i$, $s = h_{g2}$, $\mathcal{B}_i = \varnothing$;
12 $\quad\quad$ **if** $d_{gs} \geq 2r_i$ **then**
13 $\quad\quad\quad$ Continue;
14 $\quad\quad$ // $h_{g1} = g$ does not need to be considered;
15 $\quad\quad$ **for** $j = 2, \cdots, k$ **do**
16 $\quad\quad\quad$ $l = h_{gj}$ ;
17 $\quad\quad\quad$ **if** $d_{gl} >= 2r_i$ **then**
18 $\quad\quad\quad\quad$ Break;
19 $\quad\quad\quad$ **if** $\cos(\beta_i - \theta_{gl}) > d_{gl}/(2r_i)$ **then**
20 $\quad\quad\quad\quad$ Add $l$ to set $\mathcal{B}_i$
21 $\quad\quad$ Update $y_i = \arg\min_{j \in \mathcal{B}_i} \|\mathbf{x}_i - \mathbf{c}_j\|_2$;
22 Output the labels $\mathbf{y} \in \mathbb{R}^n$;

---

## 4.1 DATASETS

14 real datasets are used, including 10 middle-scale datasets: Digits, Corel5k, Coil100, CNBC, Isolet, RaFD, USPS, PINS, CPLFW, and EYaleB, as well as 4 large-scale datasets: L-CAS, L-CLBA, L-YTF, and L-EDS. The datasets contain samples ranging from 4000 to 621,126 with dimensions ranging from 256 to 1024. The statistics of the datasets are shown in Table 2, and more detailed information can be found in the appendix.

## 4.2 BASELINES

Several accelerated exact $k$-means algorithms are considered, including: Elkan's $k$-means (Elkan, 2003), Annulus $k$-means (Ann) (Drake, 2013), Exp-ns (Newling & Fleuret, 2016), and Ball $k$-means (Xia et al., 2022). To demonstrate the performance of these algorithms, standard $k$-means (Lloyd,

Table 1: Time and space complexity[1]

| Algorithms | Setup[2] | Time Cost | Space cost |
|---|---|---|---|
| Lloyd's (Lloyd, 1982) | $O(1)$ | $O(nkD)$ | $O(k+n)$ |
| Elkan's (Elkan, 2003) | $O(1)$ | $O(k^2D + knD)$ | $O(k^2 + kn)$ |
| Annulus (Drake, 2013) | $O(kD + nD)$ | $O(kD\log k + nD\log k + k^2D + knD)$ | $O(k^2 + kn)$ |
| Exp-ns (Newling & Fleuret, 2016) | $O(1)$ | $O(k^2D\log k + nD\log\log k + k^2D + knD)$ | $O(k^2 + kn)$ |
| Ball $k$-means [2] (Xia et al., 2022) | $O(k^2D + knD)$ | $O(k^2D + kmD\log m + mn'D + nD)$ | $O(k^2 + kn)$ |
| Angle $k$-means [3] | $O(nD)$ | $O(k^2D\log k + npD + nk)$ | $O(k^2)$ |

[1] $D$ is the number of features.
[2] "Setup" refers to the initialization overhead required to accelerate the algorithm.
[3] $m$ and $n'$ are the average number of cluster neighbors and samples in the active area.
[4] $p$ is the average number of candidate centers for each sample.

1982) are also considered. Since AKM does not introduce additional parameters, we only compare it with methods parameter-free.

All codes are implemented in C/C++, and python interfaces are provided. We run them on the Ubuntu machine with 64 GB main memory, 5.10 GHz i5-13600 CPU. Source code is provided in the supplementary material.

## 4.3 EXPERIMENTAL RESULTS

As can be seen from Table 3, 4, 5, and 6, and Figure 2,3, it is clearly seen that:

- On most datasets, our algorithm uses the fewest sample-center distances; and as these distance calculations dominate runtime across algorithms, it runs fastest on most datasets. Specifically, on four datasets including RaFD ($k$=200), EYaleB ($k$=200), L-YTF ($k$=2000) and L-CLBA ($k$=10000), the distance computation of our algorithm is only 10.15%, 6.7%, 15.5% and 14.81% of that of Lloyd's algorithm respectively, and the running time is only 32.13%, 39.28%, 38.39% and 52.16% of that of Lloyd's algorithm.

- When $k$ is larger for a dataset, our algorithm tends to have shorter running time and fewer distance calculations. This trend can be observed in datasets such as Corel5k, PINS, L-CLBA and L-CAS.

## 5 CONCLUSION

We introduced *Angle k-means*, a parameter-free accelerated exact $k$-means algorithm that leverages both inter-center distances and angular relationships. By precomputing these geometric properties, we developed an efficient candidate center filtering mechanism with theoretical guarantees. The algorithm maintains computational complexity competitive with distance-only methods since angles derive directly from precomputed distances. Specifically, per-iteration time complexity is $O(k^2 D \log k + npD + nk)$ with $O(k^2)$ space overhead, where $p$ is the average number of candidate centers for each sample, and $D$ is the number of the features. Experimental validation across diverse real-world datasets demonstrates significant speed improvements over state-of-the-art alternatives. The performance advantage of AKM scales with the number of clusters, becoming increasingly significant for larger values of $k$. This aligns with practical large-scale clustering scenarios where a larger sample size typically corresponds to a larger number of centers.

## 5.1 FUTURE WORK

While current angle computations reference the coordinate origin for convenience, future work will explore angular relationships relative to: 1) Multiple reference points to enhance filtering performance. 2) Dynamically selected geometric anchors. Such extensions could further reduce distance computations while maintaining theoretical guarantees.

Table 2: Description of the datasets

| Datasets | Digits | Corel5k | CNBC | Coil100 | Isolet | RaFD | USPS |
|---|---|---|---|---|---|---|---|
| $n$ | 4000 | 5000 | 6574 | 7200 | 7797 | 8040 | 9298 |
| $d$ | 256 | 423 | 256 | 1024 | 617 | 256 | 256 |
| $k$ | 10 | 50 | 236 | 100 | 26 | 67 | 10 |
| Datasets | PINS | CPLFW | EYaleB | L-CAS | L-CLBA | L-YTF | L-EDS |
| $n$ | 10770 | 11652 | 16128 | 30863 | 202599 | 621126 | 240000 |
| $d$ | 256 | 256 | 256 | 256 | 256 | 256 | 784 |
| $k$ | 100 | 3930 | 28 | 1040 | 10177 | 1595 | 10 |

Table 3: Per-iteration average of distance computation on middle-scale datasets (% of Lloyd).

| Dataset | $n$ | $d$ | $k$ | Iter | Elkan | Ann | Exp | Ball-R | Ball-noR | Angle |
|---|---|---|---|---|---|---|---|---|---|---|
| Corel5k | 5000 | 423 | 30 | 39 | 81.43% | 97.61% | 82.88% | 77.51% | 98.92% | **66.81%** |
| | | | 50 | 66 | 67.96% | 91.13% | 68.82% | 65.07% | 96.60% | **53.06%** |
| | | | 100 | 26 | 55.48% | 87.10% | 52.84% | 51.00% | 91.54% | **40.60%** |
| | | | 200 | 27 | 41.53% | 76.67% | 38.39% | 37.50% | 80.40% | **29.00%** |
| Coil100 | 7200 | 1024 | 30 | 45 | 61.80% | 72.28% | 63.75% | 58.62% | 93.61% | **47.50%** |
| | | | 50 | 39 | 51.80% | 66.57% | 51.06% | 47.89% | 87.92% | **39.71%** |
| | | | 100 | 32 | 39.23% | 56.29% | 36.83% | 35.35% | 73.81% | **29.84%** |
| | | | 200 | 29 | 25.87% | 44.15% | 22.22% | 21.57% | 57.59% | **19.50%** |
| Isolet | 7797 | 617 | 30 | 19 | 87.12% | 96.98% | 85.64% | 80.74% | 97.13% | **74.56%** |
| | | | 50 | 26 | 80.57% | 96.92% | 78.72% | 75.41% | 96.99% | **67.25%** |
| | | | 100 | 39 | 74.11% | 96.44% | 72.52% | 70.75% | 96.46% | **59.03%** |
| | | | 200 | 17 | 65.67% | 88.61% | 59.97% | 59.09% | 89.07% | **51.99%** |
| RaFD | 8040 | 256 | 30 | 16 | 64.22% | 60.80% | 57.67% | 55.82% | 62.35% | **53.71%** |
| | | | 50 | 8 | 61.02% | 58.64% | 48.78% | 46.14% | 77.51% | **42.85%** |
| | | | 100 | 11 | 42.83% | 57.42% | 34.62% | 32.91% | 64.61% | **20.51%** |
| | | | 200 | 26 | 22.28% | 50.82% | 18.92% | 18.12% | 57.19% | **10.15%** |
| PINS | 10770 | 256 | 30 | 56 | 96.37% | 99.64% | 96.52% | 93.39% | 100.12% | **83.11%** |
| | | | 50 | 25 | 94.00% | 96.32% | 90.58% | 88.50% | 96.54% | **69.79%** |
| | | | 100 | 13 | 84.96% | 91.81% | 77.51% | 76.33% | 91.99% | **43.56%** |
| | | | 200 | 20 | 75.98% | 94.65% | 71.41% | 70.62% | 94.68% | **29.93%** |
| EYaleB | 16128 | 256 | 30 | 27 | 32.60% | 46.59% | 28.11% | 22.93% | 83.48% | **15.28%** |
| | | | 50 | 31 | 21.49% | 51.36% | 20.48% | 17.03% | 89.54% | **12.42%** |
| | | | 100 | 32 | 16.95% | 51.31% | 15.10% | 13.24% | 78.15% | **10.39%** |
| | | | 200 | 59 | 10.29% | 33.06% | 9.16% | 8.24% | 57.71% | **6.70%** |

Table 4: Per-iteration average running time (s) on middle-scale datasets (% of Lloyd).

| Dataset | $n$ | $d$ | $k$ | Iter | Elkan | Ann | Exp | Ball-R | Ball-noR | Angle |
|---|---|---|---|---|---|---|---|---|---|---|
| Corel5k | 5000 | 423 | 30 | 39 | 88.51% | 4927.02% | 4831.68% | 1365.66% | 1609.40% | **83.11%** |
| | | | 50 | 66 | 86.39% | 5774.38% | 4830.50% | 1408.30% | 1319.91% | **72.60%** |
| | | | 100 | 26 | 69.70% | 6741.17% | 4415.10% | 1654.41% | 1318.73% | **54.80%** |
| | | | 200 | 27 | 60.60% | 6686.30% | 3631.00% | 1593.20% | 1506.99% | **48.88%** |
| Coil100 | 7200 | 1024 | 30 | 45 | 95.04% | 5843.50% | 6233.21% | 1801.06% | 2065.44% | **81.73%** |
| | | | 50 | 39 | 81.51% | 7948.64% | 5358.94% | 1435.50% | 1566.24% | **71.48%** |
| | | | 100 | 32 | 68.62% | 6611.39% | 4636.20% | 1353.69% | 1367.39% | **68.24%** |
| | | | 200 | 29 | 60.16% | 5972.03% | 3372.47% | 1317.36% | 1306.37% | **57.33%** |
| Isolet | 7797 | 617 | 30 | 19 | 74.43% | 4653.21% | 4821.89% | 1281.84% | 1414.29% | **67.83%** |
| | | | 50 | 26 | 68.57% | 5589.72% | 5032.31% | 1301.43% | 1241.23% | **58.75%** |
| | | | 100 | 39 | 89.58% | 9121.26% | 7070.51% | 1973.01% | 1592.17% | **78.55%** |
| | | | 200 | 17 | 82.44% | 8325.37% | 5797.87% | 2152.18% | 1542.48% | **66.97%** |
| RaFD | 8040 | 256 | 30 | 16 | 68.70% | 1995.86% | 2119.69% | 515.14% | 476.33% | **54.83%** |
| | | | 50 | 8 | 58.12% | 2202.27% | 2069.45% | 608.71% | 588.18% | **49.47%** |
| | | | 100 | 11 | 58.45% | 3477.80% | 2350.80% | 878.46% | 694.44% | **38.11%** |
| | | | 200 | 26 | 47.14% | 4380.74% | 1894.59% | 836.19% | 886.05% | **32.13%** |
| PINS | 10770 | 256 | 30 | 56 | 99.62% | 4395.61% | 4777.55% | 1069.51% | 1109.70% | **77.77%** |
| | | | 50 | 25 | 74.17% | 4096.01% | 4265.10% | 839.92% | 734.73% | **62.39%** |
| | | | 100 | 13 | 79.10% | 5369.81% | 4956.40% | 1047.10% | 791.66% | **50.68%** |
| | | | 200 | 20 | 101.48% | 9053.59% | 7421.47% | 1853.29% | 1272.49% | **56.14%** |
| EYaleB | 16128 | 256 | 30 | 27 | 50.24% | 2274.60% | 2012.81% | 836.33% | 955.72% | **41.28%** |
| | | | 50 | 31 | 50.69% | 3413.31% | 2049.89% | 848.07% | 1010.59% | **42.87%** |
| | | | 100 | 32 | 55.47% | 4867.13% | 2304.21% | 733.16% | 1131.99% | **43.28%** |
| | | | 200 | 59 | 42.96% | 4393.32% | 1882.89% | 740.65% | 999.21% | **39.28%** |

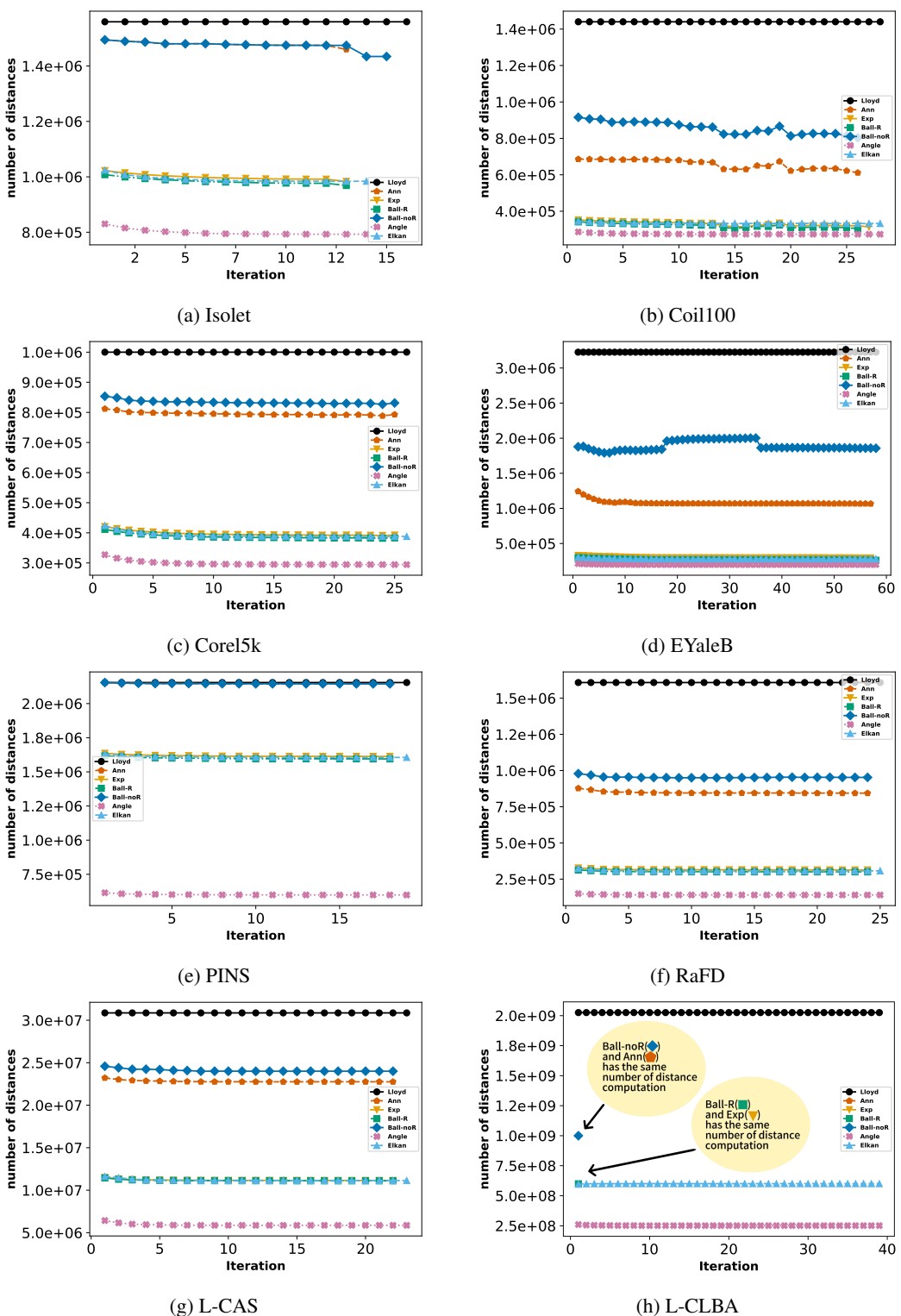

(a) Isolet

(b) Coil100

(c) Corel5k

(d) EYaleB

(e) PINS

(f) RaFD

(g) L-CAS

(h) L-CLBA

Figure 2: The number of calculated distances in each iteration on middle-scale datasets and large-scale datasets L-CAS, L-CLBA. Due to the large data volume and long computation time, only the number of distance calculations in the first iteration was recorded for Ann, Exp, Ball-R, and Ball-noR on L-CLBA.

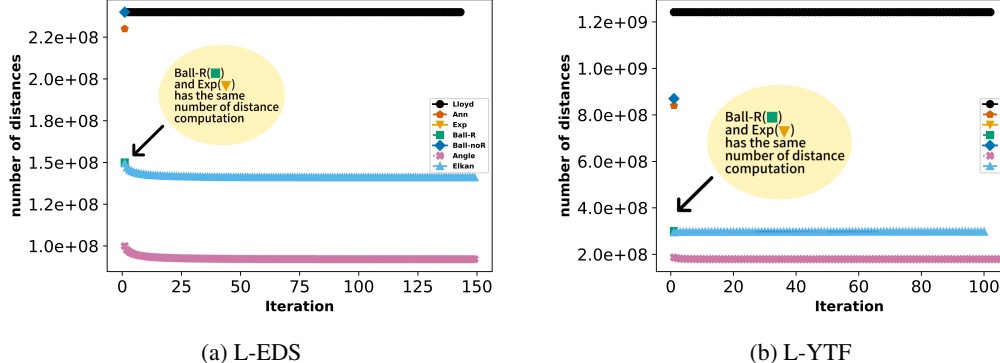

|                | (a) L-EDS |          | (b) L-YTF |

Figure 3: The number of calculated distances in each iteration on large-scale datasets L-YTF and L-EDS. Due to the large data volume and long computation time, only the number of distance calculations in the first iteration was recorded for Ann, Exp, Ball-R, and Ball-noR.

Table 5: Per-iteration average of distance computation on large-scale datasets (% of Lloyd).

| Dataset | n | d | k | Iter | Elkan | Ann | Exp | Ball-R | Ball-noR | Angle |
|---------|-----|-----|------|------|--------|-----------|-----------|-----------|-----------|--------|
| L-CAS | 30863 | 256 | 300 | 41 | 89.05% | 92.85% | 86.76% | 86.38% | 95.68% | **75.92%** |
|  |  |  | 500 | 29 | 76.51% | 86.65% | 73.14% | 72.90% | 90.27% | **57.89%** |
|  |  |  | 1000 | 24 | 38.92% | 70.92% | 34.82% | 34.67% | 74.89% | **22.71%** |
| L-CLBA | 202599 | 256 | 5000 | 72 | 41.84% | 54.75%(1) | 40.30%(1) | 40.58%(3) | 55.45%(3) | **27.70%** |
|  |  |  | 8000 | 39 | 35.23% | 52.55%(1) | 33.24%(1) | 33.22%(3) | 53.31%(3) | **18.56%** |
|  |  |  | 10000 | 40 | 31.38% | 50.45%(1) | 29.41%(1) | 29.52%(1) | 51.33%(1) | **14.81%** |
| L-EDS | 240000 | 784 | 500 | 137 | 69.58% | 97.49%(1) | 72.74%(1) | 72.41%(1) | 100.08%(1) | **48.41%** |
|  |  |  | 800 | 227 | 62.55% | 96.93%(1) | 65.84%(1) | 65.62%(1) | 100.02%(1) | **41.69%** |
|  |  |  | 1000 | 144 | 59.24% | 96.67%(1) | 62.49%(1) | 62.31%(1) | 100.00%(1) | **38.95%** |
| L-YTF | 621126 | 256 | 500 | 246 | 55.37% | 80.03%(1) | 54.04%(1) | 54.40%(5) | 82.14%(5) | **45.03%** |
|  |  |  | 1000 | 136 | 40.84% | 74.55%(1) | 39.95%(1) | 40.18%(5) | 76.30%(5) | **29.93%** |
|  |  |  | 2000 | 100 | 24.68% | 67.72%(1) | 23.82%(1) | 23.75%(1) | 70.13%(1) | **15.50%** |

[1] The numbers in parentheses indicate the iteration counts recorded for the corresponding algorithms.

Table 6: Per-iteration average running time (s) on large-scale datasets (% of Lloyd).

| Datasets | n | d | k | Iter | Elkan | Ann | Exp | Ball-R | Ball-noR | Angle |
|----------|-----|-----|------|------|---------|--------------|--------------|-------------|-------------|--------|
| L-CAS | 30863 | 256 | 300 | 41 | 115.79% | 10321.42% | 10386.03% | 1641.33% | 1419.43% | **92.93%** |
|  |  |  | 500 | 29 | 108.78% | 10589.16% | 9879.55% | 1937.89% | 1510.95% | **93.35%** |
|  |  |  | 1000 | 24 | 69.36% | 9808.12% | 5260.08% | 1995.84% | 1779.16% | **50.20%** |
| L-CLBA | 202599 | 256 | 5000 | 72 | 91.34% | 10000.17%(1) | 17011.42%(1) | 2821.78%(3) | 2189.28%(3) | **67.30%** |
|  |  |  | 8000 | 39 | 88.95% | 11965.33%(1) | 55878.09%(1) | 4705.84%(3) | 3028.96%(3) | **56.56%** |
|  |  |  | 10000 | 40 | 85.29% | 10662.26%(1) | 29391.66%(1) | 5070.91%(1) | 3301.34%(1) | **52.16%** |
| L-EDS | 240000 | 784 | 500 | 137 | 81.65% | 19795.48%(1) | 20090.79%(1) | 3613.51%(1) | 3129.86%(1) | **64.80%** |
|  |  |  | 800 | 227 | 112.51% | 33801.16%(1) | 42984.06%(1) | 5228.59%(1) | 3675.39%(1) | **85.03%** |
|  |  |  | 1000 | 144 | 104.14% | 30514.76%(1) | 44914.60%(1) | 4237.86%(1) | 3212.62%(1) | **79.41%** |
| L-YTF | 621126 | 256 | 500 | 246 | 104.14% | 13154.24%(1) | 10363.34%(1) | 1540.17%(5) | 1674.65%(5) | **91.11%** |
|  |  |  | 1000 | 136 | 81.35% | 12699.07%(1) | 9107.67%(1) | 1272.15%(5) | 1435.42%(5) | **67.17%** |
|  |  |  | 2000 | 100 | 55.19% | 11802.59%(1) | 9911.91%(1) | 958.09%(1) | 1298.81%(1) | **38.39%** |

[1] The numbers in parentheses indicate the iteration counts recorded for the corresponding algorithms.

ACKNOWLEDGMENTS

This work was supported in part by the National Natural Science Foundation of China under Grants 62406274, 32401685, and 62576304, in part by the Natural Science Foundation of Zhejiang Province, China under Grants LQN25F020029 and LQ24F020021.

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

## A    SUPPLEMENTARY MATERIAL FOR ANGLE K-MEANS

### A.1    PROOF OF LEMMA 1

**Lemma 1.** Let $\mathbf{p}$ and $\mathbf{q}$ be two points in the Euclidean space $\mathbb{R}^n$, and let $r_1$ and $r_2$ be constants greater than $0$. Then the optimal value of the following problem is $|r_1 - r_2|$:

$$\min_{\mathbf{p},\mathbf{q}} \quad \|\mathbf{p} - \mathbf{q}\|_2 \quad \text{s.t.} \quad \|\mathbf{p}\|_2 = r_1, \|\mathbf{q}\|_2 = r_2. \tag{12}$$

*Proof.* Let $\mathbf{o} = \mathbf{0}$ denote the center of the two spheres, we have $\|\mathbf{p} - \mathbf{o}\|_2 = r_1$ and $\|\mathbf{q} - \mathbf{o}\|_2 = r_2$.

$$\cos \angle \mathbf{poq} = \frac{r_1^2 + r_2^2 - \|\mathbf{p} - \mathbf{q}\|_2^2}{2 r_1 r_2} \leq 1 \Rightarrow (r_1 - r_2)^2 \leq \|\mathbf{p} - \mathbf{q}\|_2^2 \tag{13}$$

So far, it is easy to get: $\|\mathbf{p} - \mathbf{q}\|_2 \geq |r_1 - r_2|$. $\|\mathbf{p} - \mathbf{q}\|_2 = |r_1 - r_2|$ if and only if $\angle \mathbf{poq} = 0$. $\square$

### A.2    DATASETS

This section presents the sources of the datasets used. A total of 14 real datasets are used. 8 face datasets: CNBC(Gao et al., 2024), PINS, RaFD(Mahmud et al., 2018), CPLFW(Feng et al., 2020), EYaleB(Kang et al., 2015), L-CAS(Pei et al., 2023), L-CLBA(Pei et al., 2023), and L-YTF(Pei et al., 2023). 3 handwritten digits datasets: Digits, L-EDS(Jayasundara et al., 2019), and USPS(Ramirez et al., 2010). 3 other datasets: Coil100(Mohan et al., 2019), Isolet(Ramirez et al., 2010) and Corel5k(Xie et al., 2020). The datasets contain samples ranging from 4000 to 621,126 with dimensions ranging from 256 to 1024.

## A.3 EXPERIMENT

Table 7: The detailed statistics for the Angle $k$-means and Lloyd's algorithms.

| Datasets | $n$ | $Iter$ | $t_0$ | $n_0$ | Angle k-means $t_1$ | $n_2$ | $t_2$ | $t_{ak}$ | Lloyd's k-means $n_3$ | $t_3$ | $t_{km}$ |
|---|---|---|---|---|---|---|---|---|---|---|---|
| $X^{(1)}$ | 2600 | 14.34 | 0.0012 | 5300 | 0.004 | 4.4E+04 | 0.0049 | 0.1253 | 2.60E+05 | 0.0339 | 0.4863 |
| $X^{(2)}$ | 5200 | 13.54 | 0.0023 | 10500 | 0.0027 | 8.8E+04 | 0.0094 | 0.1626 | 5.20E+05 | 0.0645 | 0.8727 |
| $X^{(3)}$ | 7800 | 13.96 | 0.0033 | 15700 | 0.0022 | 1.3E+05 | 0.0147 | 0.2357 | 7.80E+05 | 0.0896 | 1.2512 |
| $X^{(4)}$ | 10400 | 14.08 | 0.0044 | 20900 | 0.0024 | 1.8E+05 | 0.0202 | 0.3207 | 1.00E+06 | 0.1056 | 1.4866 |
| $X^{(5)}$ | 13000 | 14.68 | 0.0055 | 26100 | 0.0013 | 2.2E+05 | 0.0271 | 0.4224 | 1.30E+06 | 0.1129 | 1.6573 |
| $X^{(6)}$ | 15600 | 15.04 | 0.0066 | 31300 | 0.0016 | 2.6E+05 | 0.0356 | 0.5627 | 1.60E+06 | 0.1284 | 1.9311 |
| $X^{(7)}$ | 18200 | 14.62 | 0.0078 | 36500 | 0.0012 | 3.1E+05 | 0.0439 | 0.6656 | 1.80E+06 | 0.1372 | 2.0065 |
| $X^{(8)}$ | 20800 | 14.28 | 0.0089 | 41700 | 0.0013 | 3.5E+05 | 0.0566 | 0.8338 | 2.10E+06 | 0.1561 | 2.2285 |
| $X^{(9)}$ | 23400 | 14.54 | 0.0099 | 46900 | 0.0017 | 4.0E+05 | 0.0701 | 1.0512 | 2.30E+06 | 0.1668 | 2.4253 |
| $X^{(10)}$ | 26000 | 14.62 | 0.0111 | 52100 | 0.002 | 4.4E+05 | 0.0807 | 1.2183 | 2.60E+06 | 0.1881 | 2.7496 |

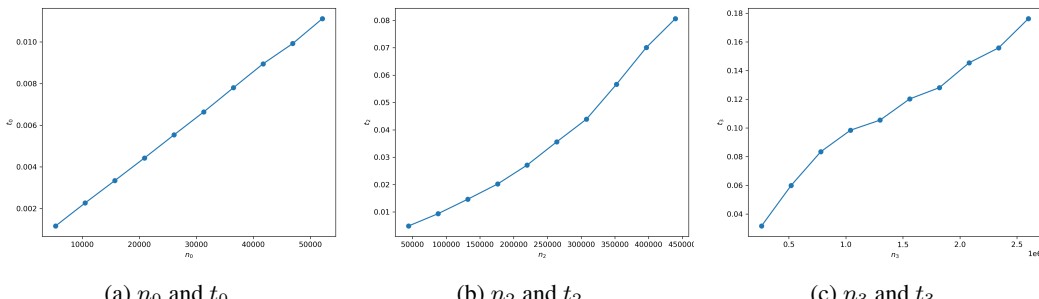

(a) $n_0$ and $t_0$      (b) $n_2$ and $t_2$      (c) $n_3$ and $t_3$

Figure 4: The relationship between the number of calculated distances and the running time.

### A.3.1 ANALYSIS OF RUNNING TIME AND DISTANCE CALCULATION RELATIONSHIP

It is difficult to find the relationship between the running time and the number of distances by looking at the results on different datasets, because the running time is related to many variables, such as the number of features, the number of clusters, and the number of iterations. Therefore, we augment the dataset $X \in \mathbb{R}^{d \times n}$ as follows to obtain $X^{(1)}, \cdots, X^{(10)}$, where $X^{(i)} = [X, \cdots, X] + e \in \mathbb{R}^{d \times in}$ where $e$ represents the noise matrix, which is used to avoid duplicate samples in $X^{(i)}$. Ar is used here, this is to say, $n = 2600$, $d = 256$, $k = 100$.

We run Lloyd's and angle $k$-means on these datasets and present the experimental results in Table 7. The meanings of the symbols involved in Tables 7 are as follows:

- $n$ is the number of samples in the dataset.
- $n_0$ is the number of distances required the initialization in Angle $k$-means.
- $t_0$ is the time required for the initialization in Angle $k$-means.
- $t_1$ is the average time spent computing auxiliary variables in each iteration.
- $n_2$ is the average number of sample-center distances in Angle $k$-means, $n_2 = np$.
- $t_2$ is the average time spent updating assignments and centers in Angle $k$-means.
- $Iter$ represents the number of iterations required for the algorithm to converge.
- $t_{ak}$ is the total time spent by Angle $k$-means, $t_{ak} = t_0 + (t_1 + t_2) \times Iter$.
- $n_3$ is the average number of sample-center distances in Lloyd's algorithm, $n_3 = nk$
- $t_3$ is the average time spent updating assignments and centers in Lloyd's algorithm.
- $t_{km}$ is the total time spent by Lloyd's algorithm, $t_{km} = t_3 \times Iter$.

From Table 7 and Figure 4, we can see that:

- In all cases, $t_1$ is much smaller than $t_2$. In other words, computing the auxiliary variable takes much less time than computing the center-sample distance.
- Since $t_{ak} = (t_2 + t_1) \times Iter + t_0$, $t_2$ occupies a major part of the total time of Angle $k$-means.
- The time used to calculate the center-sample distance, $t_2$, is approximately linearly related to the number of center-sample distances, $n_2$.

Based on the above findings, a smaller value of $p$ leads to a smaller value of $n_2$, which in turn results in a smaller running time.

### A.3.2 FULL RESULTS OF RUNNING TIME AND DISTANCE COMPUTATION

In Tables 3, 4, 5, and 6 of the article, we only show part of the results. Here are the full results. From Tables 8, 9, 10, and 11, we can see that Angle $k$-means requires fewer distance computations while maintaining faster running times on most datasets.

Table 8: Per-iteration average of distance computation on middle-scale datasets.

| Dataset | $n$ | $d$ | $k$ | Iter | Lloyd | Elkan | Ann | Exp | Ball-R | Ball-noR | Angle |
|---|---|---|---|---|---|---|---|---|---|---|---|
| Digits | 4000 | 256 | 30 | 32 | 1.2E+05 | 1.2E+05 | 1.2E+05 | 1.1E+05 | 1.1E+05 | 1.2E+05 | **1.0E+05** |
| | | | 50 | 22 | 2.0E+05 | 1.9E+05 | 1.9E+05 | 1.8E+05 | 1.7E+05 | 1.9E+05 | **1.5E+05** |
| | | | 100 | 25 | 4.0E+05 | 3.4E+05 | 3.8E+05 | 3.3E+05 | 3.2E+05 | 3.8E+05 | **2.6E+05** |
| | | | 200 | 19 | 8.0E+05 | 5.8E+05 | 7.3E+05 | 5.4E+05 | 5.3E+05 | 7.4E+05 | **4.1E+05** |
| Corel5k | 5000 | 423 | 30 | 39 | 1.5E+05 | 1.2E+05 | 1.5E+05 | 1.2E+05 | 1.2E+05 | 1.5E+05 | **1.0E+05** |
| | | | 50 | 66 | 2.5E+05 | 1.7E+05 | 2.3E+05 | 1.7E+05 | 1.6E+05 | 2.4E+05 | **1.3E+05** |
| | | | 100 | 26 | 5.0E+05 | 2.8E+05 | 4.4E+05 | 2.6E+05 | 2.5E+05 | 4.6E+05 | **2.0E+05** |
| | | | 200 | 27 | 1.0E+06 | 4.2E+05 | 7.7E+05 | 3.8E+05 | 3.7E+05 | 8.0E+05 | **2.9E+05** |
| CNBC | 6574 | 256 | 30 | 44 | 2.0E+05 | 1.5E+05 | 1.7E+05 | 1.5E+05 | 1.4E+05 | 1.8E+05 | **1.4E+05** |
| | | | 50 | 35 | 3.3E+05 | 2.4E+05 | 2.7E+05 | 2.4E+05 | 2.3E+05 | 3.0E+05 | **2.2E+05** |
| | | | 100 | 53 | 6.6E+05 | 4.1E+05 | 5.1E+05 | 4.1E+05 | 4.0E+05 | 6.2E+05 | **3.4E+05** |
| | | | 200 | 23 | 1.3E+06 | 6.0E+05 | 9.0E+05 | 5.5E+05 | 5.4E+05 | 1.1E+06 | **3.9E+05** |
| Coil100 | 7200 | 1024 | 30 | 45 | 2.2E+05 | 1.3E+05 | 1.6E+05 | 1.4E+05 | 1.3E+05 | 2.0E+05 | **1.0E+05** |
| | | | 50 | 39 | 3.6E+05 | 1.9E+05 | 2.4E+05 | 1.8E+05 | 1.7E+05 | 3.2E+05 | **1.4E+05** |
| | | | 100 | 32 | 7.2E+05 | 2.8E+05 | 4.1E+05 | 2.7E+05 | 2.5E+05 | 5.3E+05 | **2.1E+05** |
| | | | 200 | 29 | 1.4E+06 | 3.7E+05 | 6.4E+05 | 3.2E+05 | 3.1E+05 | 8.3E+05 | **2.8E+05** |
| Isolet | 7797 | 617 | 30 | 19 | 2.3E+05 | 2.0E+05 | 2.3E+05 | 2.0E+05 | 1.9E+05 | 2.3E+05 | **1.7E+05** |
| | | | 50 | 26 | 3.9E+05 | 3.1E+05 | 3.8E+05 | 3.1E+05 | 2.9E+05 | 3.8E+05 | **2.6E+05** |
| | | | 100 | 39 | 7.8E+05 | 5.8E+05 | 7.5E+05 | 5.7E+05 | 5.5E+05 | 7.5E+05 | **4.6E+05** |
| | | | 200 | 17 | 1.6E+06 | 1.0E+06 | 1.4E+06 | 9.4E+05 | 9.2E+05 | 1.4E+06 | **8.1E+05** |
| RaFD | 8040 | 256 | 30 | 16 | 2.4E+05 | 1.5E+05 | 1.5E+05 | 1.4E+05 | 1.3E+05 | 1.5E+05 | **1.3E+05** |
| | | | 50 | 8 | 4.0E+05 | 2.5E+05 | 2.4E+05 | 2.0E+05 | 1.9E+05 | 3.1E+05 | **1.7E+05** |
| | | | 100 | 11 | 8.0E+05 | 3.4E+05 | 4.6E+05 | 2.8E+05 | 2.6E+05 | 5.2E+05 | **1.6E+05** |
| | | | 200 | 26 | 1.6E+06 | 3.6E+05 | 8.2E+05 | 3.0E+05 | 2.9E+05 | 9.2E+05 | **1.6E+05** |
| USPS | 9298 | 256 | 30 | 32 | 2.8E+05 | 1.8E+05 | 2.5E+05 | 1.8E+05 | 1.7E+05 | 2.5E+05 | **1.5E+05** |
| | | | 50 | 34 | 4.6E+05 | 2.7E+05 | 4.1E+05 | 2.6E+05 | 2.5E+05 | 4.1E+05 | **2.2E+05** |
| | | | 100 | 48 | 9.3E+05 | 4.3E+05 | 8.3E+05 | 4.2E+05 | 4.0E+05 | 8.3E+05 | **3.4E+05** |
| | | | 200 | 33 | 1.9E+06 | 6.4E+05 | 1.4E+06 | 5.8E+05 | 5.7E+05 | 1.4E+06 | **5.1E+05** |
| PINS | 10770 | 256 | 30 | 56 | 3.2E+05 | 3.1E+05 | 3.2E+05 | 3.1E+05 | 3.0E+05 | 3.2E+05 | **2.7E+05** |
| | | | 50 | 25 | 5.4E+05 | 5.1E+05 | 5.2E+05 | 4.9E+05 | 4.8E+05 | 5.2E+05 | **3.8E+05** |
| | | | 100 | 13 | 1.1E+06 | 9.2E+05 | 9.9E+05 | 8.3E+05 | 8.2E+05 | 9.9E+05 | **4.7E+05** |
| | | | 200 | 20 | 2.2E+06 | 1.6E+06 | 2.0E+06 | 1.5E+06 | 1.5E+06 | 2.0E+06 | **6.4E+05** |
| CPLFW | 11652 | 256 | 30 | 37 | 3.5E+05 | 2.2E+05 | 2.3E+05 | 2.2E+05 | 2.1E+05 | 2.5E+05 | **2.0E+05** |
| | | | 50 | 48 | 5.8E+05 | 3.5E+05 | 3.7E+05 | 3.4E+05 | 3.3E+05 | 4.1E+05 | **3.2E+05** |
| | | | 100 | 44 | 1.2E+06 | 6.7E+05 | 7.1E+05 | 6.4E+05 | 6.4E+05 | 8.1E+05 | **6.2E+05** |
| | | | 200 | 53 | 2.3E+06 | 1.3E+06 | 1.3E+06 | 1.2E+06 | 1.2E+06 | 1.6E+06 | **1.2E+06** |
| EYaleB | 16128 | 256 | 30 | 27 | 4.8E+05 | 1.6E+05 | 2.3E+05 | 1.4E+05 | 1.1E+05 | 4.0E+05 | **7.4E+04** |
| | | | 50 | 31 | 8.1E+05 | 1.7E+05 | 4.1E+05 | 1.7E+05 | 1.4E+05 | 7.2E+05 | **1.0E+05** |
| | | | 100 | 32 | 1.6E+06 | 2.7E+05 | 8.3E+05 | 2.4E+05 | 2.1E+05 | 1.3E+06 | **1.7E+05** |
| | | | 200 | 59 | 3.2E+06 | 3.3E+05 | 1.1E+06 | 3.0E+05 | 2.7E+05 | 1.9E+06 | **2.2E+05** |

Table 9: Per-iteration average of distance computation on large-scale datasets.

| Datasets | $n$ | $d$ | $k$ | Iter | Lloyd | Elkan | Ann | Exp | Ball-R | Ball-noR | Angle |
|---|---|---|---|---|---|---|---|---|---|---|---|
| L-CAS | 30863 | 256 | 300 | 41 | 9.3E+06 | 8.2E+06 | 8.6E+06 | 8.0E+06 | 8.0E+06 | 8.9E+06 | **7.0E+06** |
| | | | 500 | 29 | 1.5E+07 | 1.2E+07 | 1.3E+07 | 1.1E+07 | 1.1E+07 | 1.4E+07 | **8.9E+06** |
| | | | 1000 | 24 | 3.1E+07 | 1.2E+07 | 2.2E+07 | 1.1E+07 | 1.1E+07 | 2.3E+07 | **7.0E+06** |
| L-CLBA | 202599 | 256 | 5000 | 72 | 1.0E+09 | 4.2E+08 | 5.5E+08(1) | 4.1E+08(1) | 4.1E+08(3) | 5.6E+08(3) | **2.8E+08** |
| | | | 8000 | 39 | 1.6E+09 | 5.7E+08 | 8.5E+08(1) | 5.4E+08(1) | 5.4E+08(3) | 8.6E+08(3) | **3.0E+08** |
| | | | 10000 | 40 | 2.0E+09 | 6.4E+08 | 1.0E+09(1) | 6.0E+08(1) | 6.0E+08(1) | 1.0E+09(1) | **3.0E+08** |
| L-EDS | 240000 | 784 | 500 | 137 | 1.2E+08 | 8.3E+07 | 1.2E+08(1) | 8.7E+07(1) | 8.7E+07(1) | 1.2E+08(1) | **5.8E+07** |
| | | | 800 | 227 | 1.9E+08 | 1.2E+08 | 1.9E+08(1) | 1.3E+08(1) | 1.3E+08(1) | 1.9E+08(1) | **8.0E+07** |
| | | | 1000 | 144 | 2.4E+08 | 1.4E+08 | 2.3E+08(1) | 1.5E+08(1) | 1.5E+08(1) | 2.4E+08(1) | **9.3E+07** |
| L-YTF | 621126 | 256 | 500 | 246 | 3.1E+08 | 1.7E+08 | 2.5E+08(1) | 1.7E+08(1) | 1.7E+08(5) | 2.6E+08(5) | **1.4E+08** |
| | | | 1000 | 136 | 6.2E+08 | 2.5E+08 | 4.6E+08(1) | 2.5E+08(1) | 2.5E+08(5) | 4.7E+08(5) | **1.9E+08** |
| | | | 2000 | 100 | 1.2E+09 | 3.1E+08 | 8.4E+08(1) | 3.0E+08(1) | 3.0E+08(1) | 8.7E+08(1) | **1.9E+08** |

[1] The numbers in parentheses indicate the iteration counts recorded for the corresponding algorithms.

Table 10: Per-iteration average running time (s) on middle-scale datasets

| Dataset | $n$ | $d$ | $k$ | Iter | Lloyd | Elkan | Ann | Exp | Ball-R | Ball-noR | Angle |
|---|---|---|---|---|---|---|---|---|---|---|---|
| Digits | 4000 | 256 | 30 | 32 | 3.6E-04 | 3.5E-04 | 1.5E-02 | 2.1E-02 | 4.0E-03 | 4.0E-03 | **3.2E-04** |
| | | | 50 | 22 | 4.6E-04 | 4.3E-04 | 2.5E-02 | 3.2E-02 | 6.4E-03 | 5.1E-03 | **4.2E-04** |
| | | | 100 | 25 | 7.6E-04 | 7.1E-04 | 5.0E-02 | 5.9E-02 | 1.3E-02 | 9.4E-03 | **5.8E-04** |
| | | | 200 | 19 | 1.6E-03 | 1.2E-03 | 1.0E-01 | 9.9E-02 | 2.9E-02 | 2.0E-02 | **1.0E-03** |
| Corel5k | 5000 | 423 | 30 | 39 | 6.5E-04 | 5.8E-04 | 3.2E-02 | 3.2E-02 | 8.9E-03 | 1.1E-02 | **5.4E-04** |
| | | | 50 | 66 | 8.8E-04 | 7.6E-04 | 5.1E-02 | 4.2E-02 | 1.2E-02 | 1.2E-02 | **6.4E-04** |
| | | | 100 | 26 | 1.5E-03 | 1.1E-03 | 1.0E-01 | 6.7E-02 | 2.5E-02 | 2.0E-02 | **8.3E-04** |
| | | | 200 | 27 | 2.8E-03 | 1.7E-03 | 1.9E-01 | 1.0E-01 | 4.5E-02 | 4.3E-02 | **1.4E-03** |
| CNBC | 6574 | 256 | 30 | 44 | 5.7E-04 | 4.6E-04 | 2.2E-02 | 2.0E-02 | 4.9E-03 | 8.4E-03 | **4.2E-04** |
| | | | 50 | 35 | 7.6E-04 | 7.5E-04 | 3.3E-02 | 3.6E-02 | 6.9E-03 | 8.6E-03 | **7.2E-04** |
| | | | 100 | 53 | 9.6E-04 | 9.7E-04 | 6.4E-02 | 5.6E-02 | 1.2E-02 | 1.3E-02 | **8.2E-04** |
| | | | 200 | 23 | 2.0E-03 | 1.3E-03 | 1.2E-01 | 8.1E-02 | 2.5E-02 | 2.5E-02 | **1.1E-03** |
| Coil100 | 7200 | 1024 | 30 | 45 | 2.1E-03 | 2.0E-03 | 1.2E-01 | 1.3E-01 | 3.7E-02 | 4.3E-02 | **1.7E-03** |
| | | | 50 | 39 | 2.9E-03 | 2.3E-03 | 2.3E-01 | 1.5E-01 | 4.1E-02 | 4.5E-02 | **2.1E-03** |
| | | | 100 | 32 | 4.6E-03 | 3.2E-03 | 3.0E-01 | 2.1E-01 | 6.2E-02 | 6.3E-02 | **3.1E-03** |
| | | | 200 | 29 | 8.1E-03 | 4.9E-03 | 4.9E-01 | 2.7E-01 | 1.1E-01 | 1.1E-01 | **4.7E-03** |
| Isolet | 7797 | 617 | 30 | 19 | 1.7E-03 | 1.3E-03 | 8.1E-02 | 8.4E-02 | 2.2E-02 | 2.5E-02 | **1.2E-03** |
| | | | 50 | 26 | 2.4E-03 | 1.6E-03 | 1.3E-01 | 1.2E-01 | 3.1E-02 | 2.9E-02 | **1.4E-03** |
| | | | 100 | 39 | 2.9E-03 | 2.6E-03 | 2.7E-01 | 2.1E-01 | 5.8E-02 | 4.7E-02 | **2.3E-03** |
| | | | 200 | 17 | 6.1E-03 | 5.0E-03 | 5.1E-01 | 3.6E-01 | 1.3E-01 | 9.4E-02 | **4.1E-03** |
| RaFD | 8040 | 256 | 30 | 16 | 8.4E-04 | 5.8E-04 | 1.7E-02 | 1.8E-02 | 4.3E-03 | 4.0E-03 | **4.6E-04** |
| | | | 50 | 8 | 1.2E-03 | 7.3E-04 | 2.7E-02 | 2.6E-02 | 7.6E-03 | 7.3E-03 | **6.2E-04** |
| | | | 100 | 11 | 1.6E-03 | 9.6E-04 | 5.7E-02 | 3.8E-02 | 1.4E-02 | 1.1E-02 | **6.2E-04** |
| | | | 200 | 26 | 2.5E-03 | 1.2E-03 | 1.1E-01 | 4.6E-02 | 2.1E-02 | 2.2E-02 | **7.9E-04** |
| USPS | 9298 | 256 | 30 | 32 | 9.3E-04 | 6.7E-04 | 3.1E-02 | 3.0E-02 | 9.2E-03 | 9.5E-03 | **6.1E-04** |
| | | | 50 | 34 | 9.8E-04 | 7.8E-04 | 5.1E-02 | 4.1E-02 | 1.3E-02 | 1.2E-02 | **6.4E-04** |
| | | | 100 | 48 | 1.4E-03 | 1.1E-03 | 1.0E-01 | 6.2E-02 | 2.1E-02 | 1.8E-02 | **1.0E-03** |
| | | | 200 | 33 | 2.5E-03 | 1.7E-03 | 1.9E-01 | 9.0E-02 | 3.5E-02 | 3.2E-02 | **1.5E-03** |
| PINS | 10770 | 256 | 30 | 56 | 8.3E-04 | 8.3E-04 | 3.6E-02 | 4.0E-02 | 8.9E-03 | 9.2E-03 | **6.5E-04** |
| | | | 50 | 25 | 1.5E-03 | 1.1E-03 | 6.0E-02 | 6.3E-02 | 1.2E-02 | 1.1E-02 | **9.1E-04** |
| | | | 100 | 13 | 2.2E-03 | 1.8E-03 | 1.2E-01 | 1.1E-01 | 2.3E-02 | 1.8E-02 | **1.1E-03** |
| | | | 200 | 20 | 2.9E-03 | 2.9E-03 | 2.6E-01 | 2.1E-01 | 5.3E-02 | 3.7E-02 | **1.6E-03** |
| CPLFW | 11652 | 256 | 30 | 37 | 8.9E-04 | 8.0E-04 | 3.5E-02 | 3.4E-02 | 8.7E-03 | 8.6E-03 | **7.9E-04** |
| | | | 50 | 48 | 1.2E-03 | 9.1E-04 | 5.3E-02 | 4.9E-02 | 1.1E-02 | 1.1E-02 | **7.8E-04** |
| | | | 100 | 44 | 1.8E-03 | 1.4E-03 | 1.0E-01 | 9.2E-02 | 1.6E-02 | 1.7E-02 | **1.2E-03** |
| | | | 200 | 53 | 3.0E-03 | 2.5E-03 | 1.9E-01 | 1.8E-01 | 3.0E-02 | 3.0E-02 | **2.2E-03** |
| EYaleB | 16128 | 256 | 30 | 27 | 1.6E-03 | 8.2E-04 | 3.7E-02 | 3.3E-02 | 1.4E-02 | 1.6E-02 | **6.8E-04** |
| | | | 50 | 31 | 2.0E-03 | 1.0E-03 | 6.7E-02 | 4.0E-02 | 1.7E-02 | 2.0E-02 | **8.4E-04** |
| | | | 100 | 32 | 2.7E-03 | 1.5E-03 | 1.3E-01 | 6.3E-02 | 2.0E-02 | 3.1E-02 | **1.2E-03** |
| | | | 200 | 59 | 4.1E-03 | 1.8E-03 | 1.8E-01 | 7.7E-02 | 3.0E-02 | 4.1E-02 | **1.6E-03** |

Table 11: Per-iteration average running time (s) on large-scale datasets

| Datasets | $n$ | $d$ | $k$ | Iter | Lloyd | Elkan | Ann | Exp | Ball-R | Ball-noR | Angle |
|---|---|---|---|---|---|---|---|---|---|---|---|
| L-CAS | 30863 | 256 | 300 | 41 | 1.1E-02 | 1.3E-02 | 1.1E+00 | 1.1E+00 | 1.8E-01 | 1.5E-01 | **1.0E-02** |
| | | | 500 | 29 | 1.7E-02 | 1.9E-02 | 1.8E+00 | 1.7E+00 | 3.3E-01 | 2.6E-01 | **1.6E-02** |
| | | | 1000 | 24 | 3.4E-02 | 2.3E-02 | 3.3E+00 | 1.8E+00 | 6.7E-01 | 6.0E-01 | **1.7E-02** |
| L-CLBA | 202599 | 256 | 5000 | 72 | 7.9E-01 | 7.2E-01 | 7.9E+01(1) | 1.3E+02(1) | 2.2E+01(3) | 1.7E+01(3) | **5.3E-01** |
| | | | 8000 | 39 | 1.3E+00 | 1.1E+00 | 1.5E+02(1) | 7.2E+02(1) | 6.1E+01(3) | 3.9E+01(3) | **7.3E-01** |
| | | | 10000 | 40 | 1.6E+00 | 1.4E+00 | 1.7E+02(1) | 4.8E+02(1) | 8.2E+01(1) | 5.4E+01(1) | **8.5E-01** |
| L-EDS | 240000 | 784 | 500 | 137 | 3.8E-01 | 3.1E-01 | 7.5E+01(1) | 7.6E+01(1) | 1.4E+01(1) | 1.2E+01(1) | **2.4E-01** |
| | | | 800 | 227 | 4.0E-01 | 4.5E-01 | 1.4E+02(1) | 1.7E+02(1) | 2.1E+01(1) | 1.5E+01(1) | **3.4E-01** |
| | | | 1000 | 144 | 5.3E-01 | 5.5E-01 | 1.6E+02(1) | 2.4E+02(1) | 2.2E+01(1) | 1.7E+01(1) | **4.2E-01** |
| L-YTF | 621126 | 256 | 500 | 246 | 2.5E-01 | 2.6E-01 | 3.2E+01(1) | 2.5E+01(1) | 3.8E+00(5) | 4.1E+00(5) | **2.2E-01** |
| | | | 1000 | 136 | 4.7E-01 | 3.9E-01 | 6.0E+01(1) | 4.3E+01(1) | 6.0E+00(5) | 6.8E+00(5) | **3.2E-01** |
| | | | 2000 | 100 | 9.5E-01 | 5.2E-01 | 1.1E+02(1) | 9.4E+01(1) | 9.1E+00(1) | 1.2E+01(1) | **3.6E-01** |

[1] The numbers in parentheses indicate the iteration counts recorded for the corresponding algorithms.

## A.4 THE NUMBER OF CALCULATED DISTANCES IN EACH ITERATION(RESULTS ON ALL DATASETS)

In Figure 2 and 3, we only show part of the results. Here are the full results.

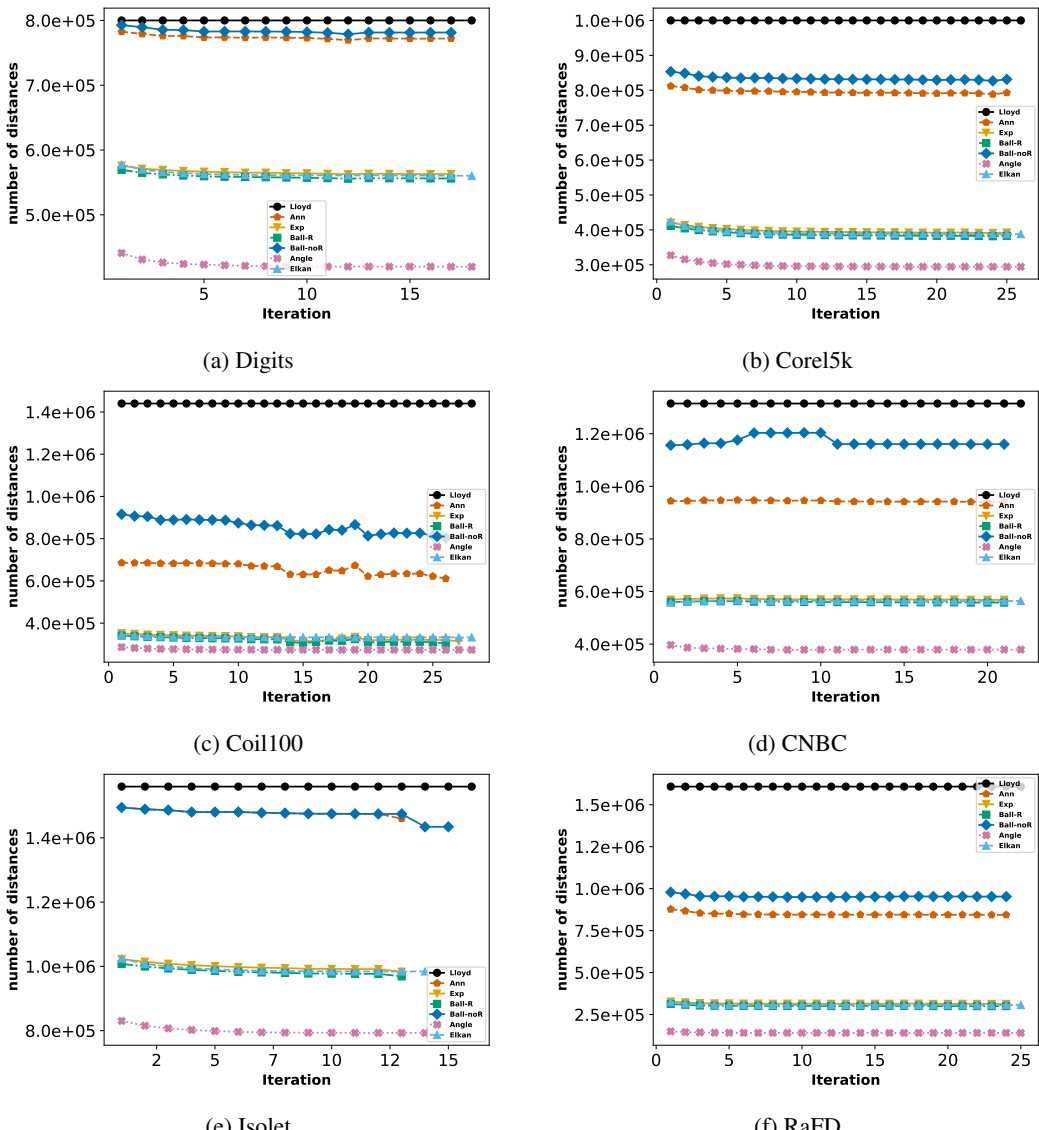

Figure 5: The number of calculated distances in each iteration on all datasets. (Part 1)

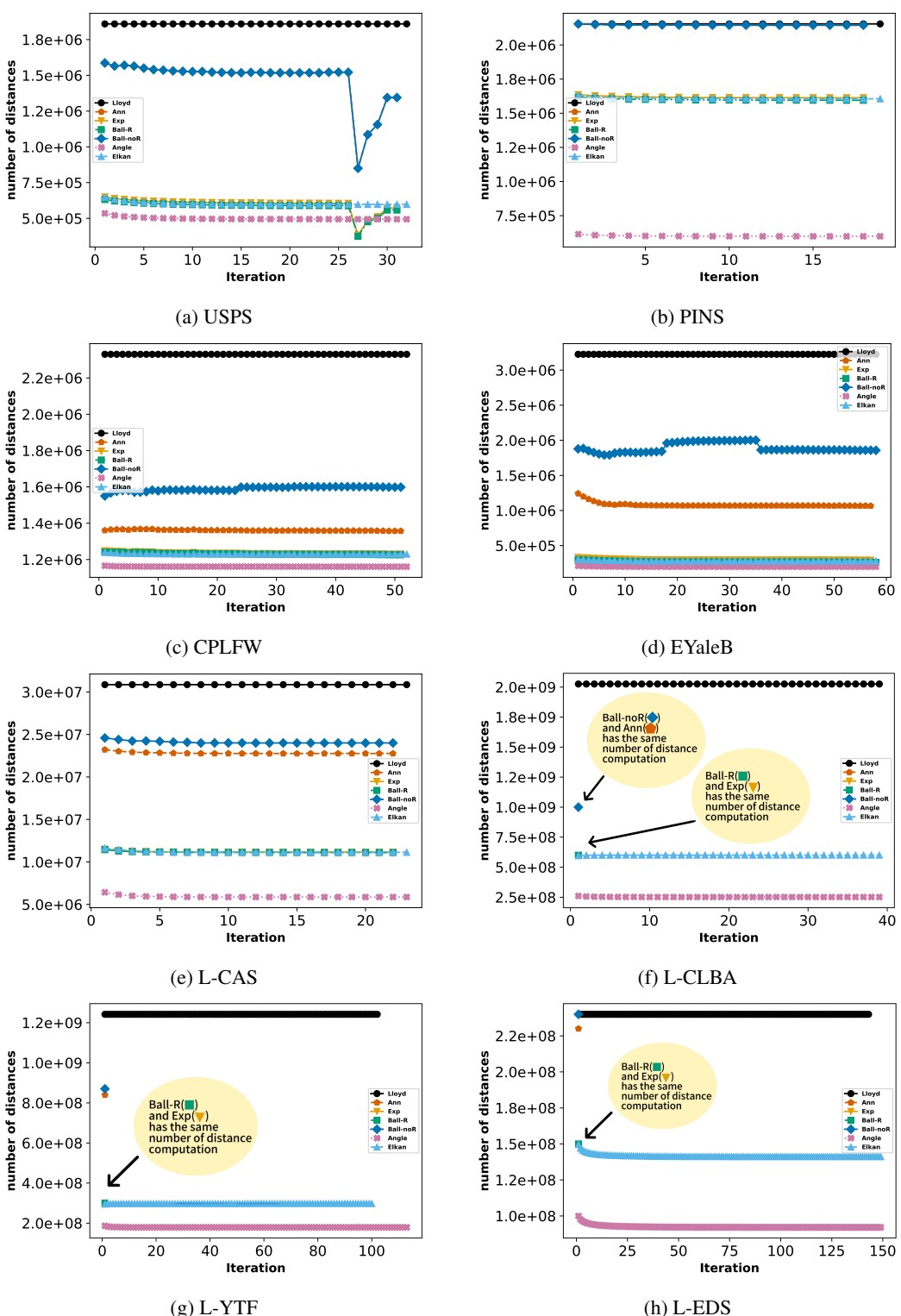

Figure 6: The number of calculated distances in each iteration on all datasets. (Part 2) (Due to the large data volume and long computation time, only the number of distance calculations in the first iteration was recorded for Ann, Exp, Ball-R, and Ball-noR on the L-CLBA, L-YTF, and L-EDS datasets.)

