# OpenReview forum: "Angle K-Means"
_ICLR.cc/2026/Conference — ICLR 2026 Poster_

### Official Review · Reviewer_joh2 · 2025-10-27

**Soundness:** 4
**Presentation:** 3
**Contribution:** 3
**Rating:** 8
**Confidence:** 4

**Summary:**

This paper provides a new way to make exact k-means faster. Consider a point, its current center, and an alternative center. The basic idea is that if the angle between the point and the alternative is large enough, and the alternative is far away from the current center, then it cannot be the a better center for the point.

**Strengths:**

This paper presents a new technique for making k-means faster. The idea is explained carefully and the experimental results are favorable.

The general idea is to calculate a score for each pair of centers, and then to use these scores to recognize for each point that many alternative centers are more distant than the point's current center. This idea (lines 175 to 177) dates back to Elkan (2003) and is illustrated in Figure 1(a) of this submission. Figure 1(c) shows that the new method can eliminate additional centers.

The new idea is a significant increment over previous research.

**Weaknesses:**

Section 2 should not just list and summarize previous methods. It should say carefully how the methods compare in time and space complexity, and in empirical speed.

There are numerous small imperfections in the English of the submission, such as a comma splice at the end of Equation 5.

What is the point of Figure 1(b)? Why not just have 1(c)? Only 1(c) is used in the actual algorithm.

Besides Lloyd's, why were Annulus, Exp-ns, and Ball hosen for comparison and not other previous algorithms? In particular, including Elkan's method would be interesting since Figure 1(c) seems to show that Angle k-means is strictly better.

Tables 3, 4, 5, 6 would be more clear if they showed other methods as a percentage of Lloyd's as a baseline.

**Questions:**

Does Figure 1(c) show that the new method is a guaranteed improvement over Elkan's method (in number of distance computations, not necessarily in clock time)? Which other methods have a similar guarantee?

Tables 3, 4, 5, 6 seem to show that speedups never exceed 10x, even with thousands of centers. What is the fundamental reason for this lack of great speedup?

In high dimensions, any two random points are almost certainly almost orthogonal, so their angle is pi/2. Is this fact relevant to the algorithm of this paper? What does it imply for how often the condition will be true in line 19 of the algorithm?

---

> ### Author Response · Authors · 2025-11-22
> **For Weaknesses 2，3，and 5**
>
> In this reply, we address the points raised in the "Weaknesses" section of your review. （Weaknesses 2， 3， and 5； Other questions will be answered later. ）
>
> We sincerely appreciate the detailed feedback and the opportunity to provide clarifications, which allow us to better explain the design and evaluation of our algorithm.
>
> - Weakness 2：In a future revision, we will remove the comma at the end of Equation 5 and carefully check for other minor language issues.
> - Weakness 3：Figure 1(b) considers two extreme cases, namely the cyan and purple areas in the figure. This illustrates that even within Elkan's candidate regions, certain points can still be pruned based on angles. This is where our idea came from, so we preserved this process.
> - Weakness 5：Regarding the presentation of the experimental data, we will take your suggestion into account in future revisions to improve clarity.

---

> ### Author Response · Authors · 2025-12-02
> **For weakness 4**
>
> Annulus, Exp-ns, and Ball K-Means are among the most representative K-Means variants, which is why we selected them.
>
> Your suggestion was very helpful, and we have added Elkan's method as a baseline. Please see Table 3-6 in the revised version.

---

> ### Author Response · Authors · 2025-12-02
> **For weakness 1**
>
> We would like to clarify that in Section 2, our intention is merely to outline the existing research routes in the field, rather than taking all these methods as comparison objects. In the experimental section of our paper, we have selected several representative algorithms and conducted detailed analyses on their theoretical complexity and actual running time.

---

### Official Review · Reviewer_qQbA · 2025-10-31

**Soundness:** 2
**Presentation:** 1
**Contribution:** 1
**Rating:** 2
**Confidence:** 4

**Summary:**

This paper presents a simple improvement to Elkan’s triangle inequality pruning method, while avoiding the need to store O(nk) distances, using the law of cosines.They give a theoretical motivation and provide experiments, comparing the average number of distance computations per iteration and the average time per iteration to other hyper-parameter free exact k-means algorithms.

**Strengths:**

The running times of the experiments look very promising.

**Weaknesses:**

Looking at the pseudo code of Algorithm 1, the loop on line 15 only stops early when the triangle inequality bound is reached (the current center is at least twice the distance from the existing one), regardless of the angle criteria. Thus the p in the running time does not refer to the average number of candidate centers per sample according to the definition of B, but rather to the average number of candidate centers under the triangle inequality bound. This gives a theoretical running time no better than what the standard triangle inequality method would give. It is not obvious how to remedy this since the angle criteria is not monotonic over centers and so cannot be used to stop early on its own. Dealing with this issue would be a significant result.
The presentation of the paper severely obfuscates the underlying idea. Lemma 1, Theorem 2, and the case distinction on page 4 appear to be completely unnecessary since equation (10) can be derived directly from the law of cosines and  (10) is the only statement required to derive the pruning rule given in the caption for Figure 1 c) that is used in Algorithm 1. Deriving (10) from the law of cosines would make the argument far simpler and easier to follow.

**Questions:**

Please clarify the discrepancy between the running time of the pseudo code and the definition of p.

Please provide an explanation for why Lemma 1, Theorem 2 and the case distinction are present.

In the abstract, it is stated that the method maintains linear time complexity wrt dimensionality, but the running times given in Section 3 are independent of dimension. Why?

What does “Setup” refer to in Table 1?

Are lines 11-13 of Algorithm 1 necessary?

Do any of the parameterised methods have sensible default parameters? It seems strange to ignore them in the experiments.

Can you explain the discrepancy between the difference in the average number of distance computations and the average running times (Table 3 vs Table 4). On some datasets (eg. Isolet, k=30), your method uses slightly fewer distance calculations, but runs over 10x faster.

---

> ### Author Response · Authors · 2025-11-13
> **Response to the First Question: Supplementary Explanation on the Time Complexity of the Proposed Algorithm.**
>
> In this reply, I'd like to address your first question.
>
> - Thank you for your careful observation and thorough reading of our paper. The comments you provided are specific and highly valuable.
> - "The loop on line 15 only stops early when the triangle inequality bound is reached, regardless of the angle criteria." That's right. However, it is important to clarify that this loop does not involve any distance calculations between samples and centers. Its function is to identify the centers that can be included in the candidate set $\mathcal{B}$.
> - Calculation of time complexity:
>   - Let $q$ denote the average number of iterations of the loop on Line 15. The time complexity of the loop is $O(nq)$ (Because the operations in Lines 16-20 can all be completed in $O(1)$ time).
>   - Let $p$ denote the average number of candidate centers per sample, and let $D$ be the feature number. The time complexity of the operation on Line 21 is $O(npD)$.
> In practical applications, $nq <<npD$, so **we did not include the cost of Line 15 ($O(nq)$) in the time complexity analysis**. Therefore, the final time complexity of our algorithm is $O(k^2Dlog k + npD)$ (the original paper stated $O(k^2log k + np)$, **because we ignored $D$**).
>   - It is worth noting that $p \leq q$. For a center $c_l$ to be included in the candidate set $\mathcal{B}$, it must not only satisfy $d_{gl} < 2r_i$ but also meet $cos(\beta_i - \theta_{gl}) > d_{gl} / (2r_i))$ — and **the latter is the key contribution** of our paper.
> - **Why did we ignore D?** In most works [1,2] related to fast K-means, the analysis of time complexity tends to focus more on the number of sample-center distance calculations. Thus, the feature dimension $D$ is often omitted. This may have caused the misunderstanding. In the revised version, we will supplement the description with the number of features $D$ to avoid ambiguity. Please feel free to leave any further questions or comments, and thank you again for your valuable comment!
>
> [1] Xia, S., Peng, D., Meng, D., Zhang, C., Wang, G., Giem, E., Wei, W., & Chen, Z. (2022). A Fast Adaptive k-means with No Bounds. IEEE Transactions on Pattern Analysis and Machine Intelligence
>
> [2] Newling, J., Fleuret, F. (2016). Fast k-means with accurate bounds. Proceedings of The 33rd International Conference on Machine Learning, in Proceedings of Machine Learning Research

---

> ### Author Response · Authors · 2025-11-22
> **Remaining issues regarding the weakness section**
>
> In this reply, we address the remaining issues raised in the Weakness section.
>
> - For the early-stopping issue, please refer to our first reply to you, *“Response to the First Question: Supplementary Explanation on the Time Complexity of the Proposed Algorithm.”*
> - Our core purpose is not to make the algorithm stop early at Line 15. The essential goal of the algorithm is to filter the candidate set $\mathcal{B}$ at Line 20 and make the size of $\mathcal{B}$ as small as possible. Even if Line 15 is executed completely, it does not affect the overall time complexity.
> - For the triangle formed by the sample point $x_i$ and cluster centers $c_j$ and $c_g$ , the following Equation can be easily derived based on the law of cosines.
>
>
>
>   $\Vert x_i - c_j \Vert^2_2 = r_i^2 + \Vert c_g - c_j \Vert^2_2 -2r_i \Vert c_g - c_j \Vert_2 cos(\angle x_i c_g c_j)$
>
>
>
>   However, it is important to note that $\angle x_i c_g c_j \not = \beta_i - \theta_{gj}$. The core reason is that the origin $O$, $x_i$, $c_g$, and $c_j$are not necessarily coplanar. If restricted to a 2D space, the law of cosines is indeed a straightforward choice; however, in high-dimensional scenarios, the core conclusion required in this paper (i.e., Theorem 2) cannot be derived solely through the law of cosines.

---

> ### Author Response · Authors · 2025-11-22
> **Response to the "Questions" Section in the Review**
>
> In this reply, I’d like to address the questions you listed in your review.
>
> 1. The pseudo code is designed to illustrate the procedure for obtaining the candidate set $\mathcal{B}$. Once $\mathcal{B}$ is determined, the number *p* — the size of this candidate set — directly determines how many distance evaluations are required for updating cluster assignments. The number of distances to be calculated can be considered as the computational complexity of the algorithm, since other operations are negligible in comparison.
> 2. Please refer to our response to the weaknesses you pointed out. **"Remaining issues regarding the weakness section"**
> 3. Our time-complexity analysis does not take the dimensionality *d* into consideration. As explained in detail in our first reply, please refer to **“Response to the First Question: Supplementary Explanation on the Time Complexity of the Proposed Algorithm.”**
> 4. “Setup” refers to the initialization overhead required to accelerate the algorithm, i.e., the preprocessing cost before the main iterations.
> 5. Yes. Lines 11-13 present a sample filtering strategy that eliminates samples whose labels do not require updates, thereby accelerating the algorithm’s runtime.
> 6. Angle K-Means adopts identical inputs to standard K-Means—specifically, only the dataset and the number of clusters c. No additional hyperparameters are required, which underscores the practicality, usability, and simplicity of our method.
> 7. For actual runtime performance, engineering optimization techniques in implementation play a non-negligible role. For a detailed discussion, refer to our response to Reviewer 5Q16 please (*“The Impact of Programming Implementations on Runtime Performance.”*)

---

> > ### Comment · Reviewer_qQbA · 2025-11-25
> >
> > Thank you for your clarifications.
> >
> > Running time: Thank you for the explanation. I think the nuance surrounding nq vs npD should absolutely be explained somewhere in the paper. In worst case scenarios, q may be O(k) even if p is much smaller. This has the effect of hiding an extra O(nk) term in the running time. Not only do we need p to be small, we also need q to be too. Obviously in high dimensional spaces this will still look favourable compared to other factors of O(nk) in the other algorithms which pick up the D term, but it is still important.
> >
> > Law of cosines: I’m still confused on this point. As you mentioned, equation (10) holds by the law of cosines if we replace $\beta_i - \theta_{gj}$ with $\angle x_i c_g c_j$. How can it simultaneously hold with $\beta_i - \theta_{gj}$ instead, unless $\beta_i - \theta_{gj} = \angle x_i c_g c_j$, at least up to the sign.

---

> > > ### Author Response · Authors · 2025-11-25
> > >
> > > Thank you for your feedback.
> > >
> > > **Regarding the running time:** We will incorporate Dimension D into the revised version to enhance the clarity of the expression.
> > >
> > > **Regarding the Eq.(10):**
> > > - Eq.(10) is derived from Theorem 2 rather than the Law of Cosines.
> > > - $\beta_i - \theta_{gj} \not = \angle x_i c_g c_j$.
> > > - According to Theorem 2, we have:
> > > $d(p, q) = \sqrt{e_2^2 + e_4^2 - 2e_2 e_4 cos(\theta_1 - \theta_2)}$.
> > > By substituting $\mathbf{p}$ with $\mathbf{x}_i$, $\mathbf{q}$ with $\mathbf{c}_j$, $\mathbf{a}$ with $\mathbf{O}$,
> > > and $\mathbf{b}$ with $\mathbf{c}_g$
> > > we have
> > > $e_2^2 = r_i^2$,
> > > $e_4^2 = \Vert c_i - c_g \Vert_2^2$
> > > $\theta_1 = \beta_i$
> > > $\theta _2 = \theta _{gj} $
> > > Thus, we can obtain the desired conclusion.

---

> > > ### Author Response · Authors · 2025-11-25
> > >
> > > If our explanation remains unclear, please do not hesitate to leave further comments.

---

> > > ### Author Response · Authors · 2025-11-26
> > > **Regarding the running time**
> > >
> > > To avoid introducing new notation, we directly denote the complexity of the loop in Line 15 as $O(nk)$, and have revised Table 1 accordingly.
> > >
> > > Table 1: Time and space complexity
> > >
> > > | Algorithms                          | Setup      | Worst-case                                      | Space cost   |
> > > | :--------------------------------- | :--------- | :---------------------------------------------- | :----------- |
> > > | Lloyd's (Lloyd, 1982)                       | $O(1)       $     | $O(nkD) $                                                       | $O(k + n) $|
> > > | Annulus (Drake, 2013)                    | $O((k + n)D)$      | $O(kD\log k + nD\log k + k^2D + knD)        $   | $O(k^2 + kn) $|
> > > | Exp-ns (Newling & Fleuret, 2016)   | $O(1) $           | $O(k^2D\log k + nD\log\log k + k^2D + knD)$  | $O(k^2 + kn) $|
> > > | Ball k-means (Xia et al., 2022)        | $O((k^2 + kn)D)$ | $O(k^2D + kmD\log m + mn'D + nD)   $           | $O(k^2 + kn) $|
> > > | Angle k-means                                | $O(nD)$             | $O(k^2 D\log k + npD + nk) $                                   | $O(k^2) $  |
> > >
> > > Compared with Annulus and Exp-ns, the $O(nk)$ term involved in the complexity of our algorithm is far smaller than the $O(nkD)$ term involved in Annulus and Exp-ns. The core reason lies in that the loop in Line 15 of our algorithm does not involve distance calculation. It should be noted that the value of D (dimension) is usually several hundred, or even several thousand in practical scenarios, which further amplifies the complexity advantage of our algorithm.

---

> > > > ### Comment · Reviewer_qQbA · 2025-11-26
> > > >
> > > > Thanks for your comments. I am satisfied with the running time now. Regarding Equation (10):
> > > >
> > > > My question is as follows. From my understanding, from the law of cosines (not Theorem 2), it follows that
> > > >
> > > > $$\|x_i - c_j\|_2^2= r_i^2 + \|c_g - c_j\|_2^2 - 2 r_i \|c_g - c_j\|_2 \cos(\gamma)$$
> > > >
> > > >
> > > > where $\gamma = \angle x_i c_g c_j$ is the interior angle at $c_g$, defined in the 2-dimensional plane spanned by $u = x_i - c_g$, and  $v = c_j - c_g$. Equation (10) is written as:
> > > >
> > > > $$\|x_i - c_j\|2^2= r_i^2 + \|c_g - c_j\|_2^2- 2 r_i \|c_g - c_j\|_2 \cos(\beta_i - \theta{gj})$$
> > > >
> > > > For these both to hold, we must have $\cos(\gamma) = \cos(\beta_i - \theta_{gj})$. What am I missing?

---

> > > > > ### Author Response · Authors · 2025-11-26
> > > > >
> > > > > First, we revisit the content of the paper:
> > > > >
> > > > > ==================(Page 4)
> > > > >
> > > > > From Theorem 2, we know that the square of the $\textcolor{red}{\text{smallest Euclidean distance}}$ between $x_i$ and $c_j$ is
> > > > > $\Vert x_i − c_j \Vert_2^2 = r^2_i + \Vert c_g − c_j \Vert_2^2 − 2r_i \Vert c_g − c_j \Vert_2  cos(\beta_i − \theta_{gj}) $ ------ (10)
> > > > >
> > > > > ==================
> > > > >
> > > > > In other words, our Equation 10 aims to find the minimum value of $\Vert x_i - c_j \Vert_2^2$, which is a crucial point.
> > > > >
> > > > > ---
> > > > > Second, according to the Law of Cosines, we have:
> > > > >
> > > > > $\Vert x_i - c_j \Vert_2^2= r_i^2 + \Vert c_g - c_j\Vert_2^2 - 2 r_i \Vert c_g - c_j\Vert_2 \cos(\gamma)$
> > > > >
> > > > > where $\gamma = \angle x_i c_g c_j$.
> > > > >
> > > > > To find the minimum value of $\Vert x_i - c_j \Vert_2^2$ (based on the Law of Cosines), (both $r_i$ and $\Vert c_g - c_j\Vert_2^2$ are fixed), we are essentially minimizing $\gamma$.
> > > > >
> > > > > ---
> > > > > Finally, combining Theorem 2 and the Law of Cosines, we know:
> > > > >
> > > > > $ \beta_i − \theta_{gj} \leq \gamma$.
> > > > >
> > > > > Therefore, **your point is correct**. However, our focus is on minimizing the value of $\Vert x_i - c_j \Vert_2^2$ rather than expressing it by $\angle x_i c_g c_j$.
> > > > >
> > > > > ---
> > > > > We apologize for any confusion caused, as the Law of Cosines was not explicitly considered in the paper; instead, the minimum distance formula between two points was directly derived （Theorem 2）.
> > > > >
> > > > > Directly applying the Law of Cosines would involve the angle $\angle x_i c_gc_j$, but the computational cost of calculating this angle is comparable to that of directly computing the distance from $x_i$ to $c_j$. Therefore, we did not adopt the Law of Cosines to accelerate the K-means algorithm.

---

> > > > > ### Author Response · Authors · 2025-11-26
> > > > >
> > > > > If our explanation remains unclear, please do not hesitate to leave further comments.

---

### Official Review · Reviewer_NsYa · 2025-10-31

**Soundness:** 3
**Presentation:** 3
**Contribution:** 2
**Rating:** 4
**Confidence:** 4

**Summary:**

The paper introduces Angle k-means, an exact and parameter-free acceleration of standard k-means that exploits geometric angle relations between cluster centers to skip redundant distance calculations. By precomputing center distances and angles, it preserves identical results to Lloyd’s k-means while greatly improving efficiency. Compared with Ball k-means and Exponion k-means, it achieves faster clustering without accuracy loss. Theoretical analysis shows linear complexity in data size, and experiments on 14 datasets report up to 90% fewer distance computations.

**Strengths:**

The algorithm relies on clear geometric intuition (angle constraints) without complex modifications.

Achieves up to 80–90% reduction in distance computations on real datasets.

Well-suited for low- and mid-dimensional large-scale datasets in industrial use.

**Weaknesses:**

The idea is largely derived from existing geometric pruning strategies, with only minor reformulation via angular constraints.

The approach strongly resembles recent TILB k-means methods that also exploit triangle-inequality-based geometric filtering, yet the paper provides no comparison or discussion.

The algorithm only applies to Euclidean metrics and cannot extend to cosine or Mahalanobis distances.

Angle-based pruning loses effectiveness in high-dimensional spaces.

**Questions:**

See weakness

---

> ### Author Response · Authors · 2025-11-21
> **We believe the reviewer may not possess adequate familiarity with this specific research domain.**
>
> We kindly request the Area Chairs (ACs) to carefully consider the comments provided by Reviewer NsYa. Although the reviewer’s confidence score is 4, we respectfully submit that there may be a lack of familiarity with fast k-means clustering
>
> - TILB K-means is not a clustering algorithm, and therefore cannot be included as a comparative method.
> - Our core filtering formula required over a page of rigorous proof, which the reviewer dismissively called a "minor modification." This characterization is entirely unacceptable.
> - Our algorithm maintains strong acceleration performance even in high-dimensional settings, and corresponding experimental results have been added to the Comment.
> - Criticizing our Euclidean-based method for not extending to other distance metrics would logically disqualify nearly all prior work in this well-established subfield.

---

> ### Author Response · Authors · 2025-11-22
> **Response to the comment on TILB k-means.**
>
> In this reply, I'd like to address your second question.
>
> Thank you for investing your time in reviewing our work. We are grateful for your detailed observations and constructive suggestions, which provide meaningful directions for improvement.
>
> K-means clustering consists of two distinct stages:
>
> 1. Selecting initial centers
> 2. iteratively reassigning points and updating centers.
>
> The TILB k-means[1] methods are designed to accelerate k-means++ seeding (KMS), i.e., step 1, by reducing the number of distance computations when choosing initial centers. In contrast, our Angle k-means algorithm focuses on accelerating the reassignment step, i.e., step 2. Therefore, TILB and our method target different parts of the k-means procedure, and a direct experimental comparison is not appropriate.
>
> [1] Zhang, H., & Li, J. (2025). Towards faster seeding for k-means++ via lower bound and triangle inequality. *Neurocomputing*, 639, 130227.

---

> ### Author Response · Authors · 2025-11-22
> **Effectiveness of Angle-Based Pruning in High-Dimensional Spaces**
>
> In this reply, I'd like to address your fourth question.
>
> The phenomenon that the inner product of two random vectors in high-dimensional space tends to zero has little to do with our algorithm. The core reason is that our formula is not designed for arbitrary random vectors, but specifically targets the triangular structure formed by the sample point $x_i$ and cluster centers $c_g$ and $c_j$.
>
> We reorganized the numerical results from Tables 3, 4, 5, and 6 into a new summary table, selecting representative datasets from these tables.
>
> - Table 1: Per-iteration average number of distance computations for different values of *d*.
>
>   | Dataset | n     | d    | k   | Iter | Ann/Lloyd | Exp/Lloyd | Ball-R/Lloyd | Ball-noR/Lloyd | Angle /Lloyd |
>   | ------- | ----- | ---- | --- | ---- | --------- | --------- | ------------ | -------------- | ------------ |
>   | EYaleB  | 16128 | 256  | 30  | 27   | 46.60%    | 28.10%    | 22.90%       | 83.50%         | 15.30%       |
>   | Isolet  | 7797  | 617  | 30  | 19   | 97.00%    | 85.60%    | 80.70%       | 97.10%         | 74.60%       |
>   | Coil100 | 7200  | 1024 | 30  | 45   | 72.30%    | 63.80%    | 58.60%       | 93.60%         | 47.50%       |
>   | EYaleB  | 16128 | 256  | 50  | 31   | 51.40%    | 20.50%    | 17.00%       | 89.50%         | 12.40%       |
>   | Isolet  | 7797  | 617  | 50  | 26   | 96.90%    | 78.70%    | 75.40%       | 97.00%         | 67.20%       |
>   | Coil100 | 7200  | 1024 | 50  | 39   | 66.60%    | 51.10%    | 47.90%       | 87.90%         | 39.70%       |
>   | EYaleB  | 16128 | 256  | 100 | 32   | 51.30%    | 15.10%    | 13.20%       | 78.10%         | 10.40%       |
>   | Isolet  | 7797  | 617  | 100 | 39   | 96.40%    | 72.50%    | 70.70%       | 96.50%         | 59.50%       |
>   | Coil100 | 7200  | 1024 | 100 | 32   | 56.30%    | 36.80%    | 35.40%       | 73.80%         | 29.80%       |
>   | EYaleB  | 16128 | 256  | 200 | 59   | 33.10%    | 9.20%     | 8.20%        | 57.70%         | 6.70%        |
>   | Isolet  | 7797  | 617  | 200 | 17   | 88.60%    | 60.00%    | 59.10%       | 89.10%         | 52.40%       |
>   | Coil100 | 7200  | 1024 | 200 | 29   | 44.10%    | 22.20%    | 21.60%       | 57.60%         | 19.50%       |
>
> - Table 2: Per-iteration average running time (s) for different values of *d*
>
>   | Dataset | n     | d    | k   | Iter | Ann/Lloyd | Exp/Lloyd | Ball-R/Lloyd | Ball-noR/Lloyd | Angle /Lloyd |
>   | ------- | ----- | ---- | --- | ---- | --------- | --------- | ------------ | -------------- | ------------ |
>   | EYaleB  | 16128 | 256  | 30  | 27   | 2274.60%  | 2012.80%  | 836.30%      | 955.70%        | 41.30%       |
>   | Isolet  | 7797  | 617  | 30  | 19   | 4653.20%  | 4821.90%  | 1281.80%     | 1414.30%       | 67.80%       |
>   | Coil100 | 7200  | 1024 | 30  | 45   | 5843.50%  | 6233.20%  | 1801.10%     | 2065.40%       | 81.70%       |
>   | EYaleB  | 16128 | 256  | 50  | 31   | 3413.30%  | 2049.90%  | 848.10%      | 1010.60%       | 42.90%       |
>   | Isolet  | 7797  | 617  | 50  | 26   | 5589.70%  | 5032.30%  | 1301.40%     | 1241.20%       | 58.80%       |
>   | Coil100 | 7200  | 1024 | 50  | 39   | 7948.60%  | 5358.90%  | 1435.50%     | 1566.20%       | 71.50%       |
>   | EYaleB  | 16128 | 256  | 100 | 32   | 4867.10%  | 2304.20%  | 733.20%      | 1132.00%       | 43.30%       |
>   | Isolet  | 7797  | 617  | 100 | 39   | 9121.30%  | 7070.50%  | 1973.00%     | 1592.20%       | 78.60%       |
>   | Coil100 | 7200  | 1024 | 100 | 32   | 6611.40%  | 4636.20%  | 1353.70%     | 1367.40%       | 68.20%       |
>   | EYaleB  | 16128 | 256  | 200 | 59   | 4393.30%  | 1882.90%  | 740.60%      | 999.20%        | 39.30%       |
>   | Isolet  | 7797  | 617  | 200 | 17   | 8325.40%  | 5797.90%  | 2152.20%     | 1542.50%       | 67.00%       |
>   | Coil100 | 7200  | 1024 | 200 | 29   | 5972.00%  | 3372.50%  | 1317.40%     | 1306.40%       | 57.30%       |
>
> From Tables 1 and 2, we observe that:
>
> - Angle K-Means consistently maintains clear advantages over Lloyd's K-Means across all dimensionalities evaluated, with this advantage being particularly evident on datasets such as EYaleB.

---

> > ### Author Response · Authors · 2025-11-22
> > **Additional Experiments on High-Dimensional Datasets**
> >
> > To further strengthen our evidence, we conducted new experiments on additional datasets.
> >
> > - Table 3: Per-iteration average number of distance computations for different values of *d* on additional datasets
> >
> >   | Dataset | n     | d     | k   | Iter | Ann/Lloyd | Exp/Lloyd | Ball-R/Lloyd | Ball-noR/Lloyd | Angle/Lloyd |
> >   | ------- | ----- | ----- | --- | ---- | --------- | --------- | ------------ | -------------- | ----------- |
> >   | CMUPIE  | 2856  | 1024  | 100 | 26   | 30.9%     | 21.6%     | 20.1%        | 41.0%          | 18.2%       |
> >   | STL10   | 13000 | 2048  | 200 | 36   | 97.2%     | 92.4%     | 91.7%        | 97.4%          | 85.9%       |
> >   | Mpeg7   | 1400  | 6000  | 100 | 9    | 81.5%     | 69.5%     | 68.4%        | 81.5%          | 67.7%       |
> >   | Face94  | 2640  | 36000 | 10  | 36   | 89.3%     | 96.9%     | 84.6%        | 102.8%         | 66.2%       |
> >
> > - Table 4: Per-iteration average running time (s) for different values of *d* on additional datasets
> >
> >   | Dataset | n     | d     | k   | Iter | Ann/Lloyd | Exp/Lloyd | Ball-R/Lloyd | Ball-noR/Lloyd | Angle/Lloyd |
> >   | ------- | ----- | ----- | --- | ---- | --------- | --------- | ------------ | -------------- | ----------- |
> >   | CMUPIE  | 2856  | 1024  | 100 | 26   | 2829.5%   | 2099.2%   | 1144.3%      | 888.6%         | 39.7%       |
> >   | STL10   | 13000 | 2048  | 200 | 36   | 45504.6%  | 59353.4%  | 2610.2%      | 2053.0%        | 75.4%       |
> >   | Mpeg7   | 1400  | 6000  | 100 | 9    | 22723.8%  | 24511.2%  | 2786.2%      | 1890.5%        | 82.0%       |
> >   | Face94  | 2640  | 36000 | 10  | 36   | 4346.7%   | 6394.3%   | 1678.4%      | 1720.6%        | 52.1%       |
> >
> > These new results validate that Angle K-Means continues to achieve consistent improvements in both computation cost and runtime across various dimensionality levels and data distributions.

---

### Official Review · Reviewer_5Q16 · 2025-10-31

**Soundness:** 3
**Presentation:** 3
**Contribution:** 3
**Rating:** 6
**Confidence:** 3

**Summary:**

The paper proposes an angle-based acceleration approach for the K-means clustering algorithm, referred to as Angle-means. The main idea is to use angular bounds to prune unlikely candidate centers in the assignment step, thereby reducing the number of distance computations. The authors derive a theoretical bound (Theorem 2) based on geometric analysis and compare Angle-means with several existing acceleration variants, including Annulus, Exp-ns, and Ball K-means. Experimental results show that Angle-means achieves comparable clustering accuracy but faster runtime than baseline methods on multiple datasets.

**Strengths:**

1. Solid geometric reasoning behind the angular pruning criterion.

2. Simple algorithmic design that is easy to implement and integrate with existing K-means frameworks.

3. Empirical results demonstrate consistent, though moderate, runtime gains compared to several baseline acceleration methods.

4. Theoretical guarantees (Lemma 1, Theorem 2) are sound and clearly derived.

**Weaknesses:**

1. The proposed algorithm is an acceleration method for exact K-means, but lacks comparison with approximate K-means algorithms (e.g., uniform sampling, coreset-based methods) in terms of runtime, clustering cost, and accuracy (ACC). Such comparisons would help clarify the trade-off between precision and efficiency.

2. While the geometric intuition is sound, the theoretical analysis remains relatively elementary. A more rigorous convergence analysis or worst-case guarantee (beyond per-iteration complexity) would significantly strengthen the contribution. In particular, it would be useful to discuss whether the parameter p in the time complexity can be expressed or bounded in expectation as a function of 𝑘.

3. When the number of clusters 𝑘 is very large, the proposed acceleration scheme seems to offer limited benefit, as the overhead of managing angular relationships may offset the gain from pruning.

**Questions:**

1. The paper evaluates five algorithms: Lloyd, Annulus, Exp-ns, Ball K-means, and Angle-means. The latter four are intended as accelerated versions of Lloyd. According to Table 3, all of them reduce the number of distance computations compared to Lloyd. However, in Table 4 (runtime), only Angle-means runs faster, while Annulus, Exp-ns, and Ball K-means are actually slower. Could the authors clarify why this happens? Is it mainly due to additional preprocessing or bookkeeping costs?

---

> ### Author Response · Authors · 2025-11-15
> **The Impact of Programming Implementation on Runtime (For questions not weakness)**
>
> First of all, thank you for your feedback. In this reply, I'd like to address your question (it's not for the section of weakness).
>
> - The actual running time of the K-means is not solely determined by the mathematical optimization algorithm, engineering techniques in programming also play a non-negligible role. For example, in the case of matrix multiplication $AB=C$, approaches that leverage mechanisms such as instruction set optimization and cache hit optimization can significantly outperform naive triple nested loop traversal in terms of efficiency.
>
> - Lloyd's algorithm adopted in this paper is implemented by scikit-learn$^1$ (sklearn). As a widely used high-performance machine learning toolkit, the K-means module of sklearn incorporates extensive engineering-level optimization strategies (e.g., vectorized operations, memory layout optimization, etc.). Therefore, even though the theoretical computational complexity of Lloyd‘s algorithm is higher than those accelerated algorithms, it still maintains fast runtime.
>
> - This paper not only proposes an optimization algorithm but also provides an efficient implementation. Our code has been submitted as supplementary material, which can verify the efficiency of its implementation—this is also a key reason why the proposed algorithm can outperform the Lloyd algorithm in terms of running time.
>
> $^1$ https://scikit-learn.org/

---

> ### Author Response · Authors · 2025-11-21
> **Weakness 3: Clarification on the Overall Complexity and Advantage of Our Algorithm for Large $k$**
>
> In this reply, I'd like to address the third weakness you pointed out.
>
> The computational complexity of standard k-means is $O(nk)$. Even with the addition of an $O(k^2)$ overhead,
> the overall complexity $O((n+k)k)$ remains nearly equivalent to $O(nk)$—this is because in practical clustering tasks, the number of samples $n$ is typically much larger than the number of clusters $k$ (i.e., $n \gg k)$, so **the impact of the $k^2$ term on the overall complexity is negligible**.
>
> Notably, the advantage of our algorithm becomes more pronounced when the number of clusters $k$ is large. The core reason is that our algorithm is designed to reduce the matching cost between samples and cluster centers through efficient center screening. As $k$ increases, the potential proportion of invalid matches rises, and the optimization space of our algorithm in the center screening stage becomes larger—thereby further highlighting its acceleration effect.
>
> To illustrate the effect of large $k$ on acceleration performance, we **reorganized a subset of the original experimental results** and provide the following table:
>
> Table 1: Per-iteration average number of distance computations for different values of $k$
>
> | Dataset | $n$    | $d$    | $k$   | Iter | Ann/Lloyd | Exp/Lloyd | Ball-R/Lloyd | Ball-noR/Lloyd | Angle/Lloyd |
> | ------- | ---- | ---- | --- | ---- | --------- | --------- | ------------ | -------------- | ----------- |
> | RaFD    | 8040 | 256  | 30  | 16   | 60.8%     | 57.7%     | 55.8%        | 62.4%          | 53.7%       |
> |         |      |      | 50  | 8    | 58.6%     | 48.8%     | 46.1%        | 77.5%          | 42.8%       |
> |         |      |      | 100 | 11   | 57.4%     | 34.6%     | 32.9%        | 64.6%          | 20.5%       |
> |         |      |      | 200 | 26   | 50.8%     | 18.9%     | 18.1%        | 57.2%          | 10.2%       |
> | Coil100 | 7200 | 1024 | 30  | 45   | 72.3%     | 63.8%     | 58.6%        | 93.6%          | 47.5%       |
> |         |      |      | 50  | 39   | 66.6%     | 51.1%     | 47.9%        | 87.9%          | 39.7%       |
> |         |      |      | 100 | 32   | 56.3%     | 36.8%     | 35.4%        | 73.8%          | 29.8%       |
> |         |      |      | 200 | 29   | 44.1%     | 22.2%     | 21.6%        | 57.6%          | 19.5%       |
> | Corel5k | 5000 | 423  | 30  | 39   | 97.6%     | 82.9%     | 77.5%        | 98.9%          | 66.8%       |
> |         |      |      | 50  | 66   | 91.1%     | 68.8%     | 65.1%        | 96.6%          | 53.1%       |
> |         |      |      | 100 | 26   | 87.1%     | 52.8%     | 51.0%        | 91.5%          | 40.6%       |
> |         |      |      | 200 | 27   | 76.7%     | 38.4%     | 37.5%        | 80.4%          | 29.0%       |
>
> Table 2: Per-iteration average running time (s) for different values of $k$
>
> | Dataset | $n$    |  $d$    | $k$   | Iter | Ann/Lloyd | Exp/Lloyd | Ball-R/Lloyd | Ball-noR/Lloyd | Angle/Lloyd |
> | ------- | ---- | ---- | --- | ---- | --------- | --------- | ------------ | -------------- | ----------- |
> | RaFD    | 8040 | 256  | 30  | 16   | 1995.9%   | 2119.7%   | 515.1%       | 476.3%         | 54.8%       |
> |         |      |      | 50  | 8    | 2202.3%   | 2069.4%   | 608.7%       | 588.2%         | 49.5%       |
> |         |      |      | 100 | 11   | 3477.8%   | 2350.8%   | 878.5%       | 694.4%         | 38.1%       |
> |         |      |      | 200 | 26   | 4380.7%   | 1894.6%   | 836.2%       | 886.0%         | 32.1%       |
> | Coil100 | 7200 | 1024 | 30  | 45   | 5843.5%   | 6233.2%   | 1801.1%      | 2065.4%        | 81.7%       |
> |         |      |      | 50  | 39   | 7948.6%   | 5358.9%   | 1435.5%      | 1566.2%        | 71.5%       |
> |         |      |      | 100 | 32   | 6611.4%   | 4636.2%   | 1353.7%      | 1367.4%        | 68.2%       |
> |         |      |      | 200 | 29   | 5972.0%   | 3372.5%   | 1317.4%      | 1306.4%        | 57.3%       |
> | Corel5k | 5000 | 423  | 30  | 39   | 4927.0%   | 4831.7%   | 1365.7%      | 1609.4%        | 83.1%       |
> |         |      |      | 50  | 66   | 5774.4%   | 4830.5%   | 1408.3%      | 1319.9%        | 72.6%       |
> |         |      |      | 100 | 26   | 6741.2%   | 4415.1%   | 1654.4%      | 1318.7%        | 54.8%       |
> |         |      |      | 200 | 27   | 6686.3%   | 3631.0%   | 1593.2%      | 1507.0%        | 48.9%       |
>
> From these experimental results, we observe that the proposed method does not exhibit performance degradation when $k$ increases. Instead, both the number of distance computations and the running time improve as $k$ becomes larger.

---

> ### Comment · Reviewer_5Q16 · 2025-11-27
> **Response to the Authors**
>
> Thanks for the clarifications. I prefer to maintain my initial evaluations at this stage.

---

> > ### Author Response · Authors · 2025-11-27
> > **Response to the Reviewer**
> >
> > Thank you for your comment. We are currently working on the rigorous bound for parameter $p$ with respect to $k$ and will complete it before December 1. We will update our response accordingly and appreciate your further review.

---

> ### Author Response · Authors · 2025-12-02
> **Weakness 1**
>
> This request is respectfully declined. Our work specifically accelerates exact k-means, guaranteeing identical results to Lloyd's algorithm. Approximate methods, which trade accuracy for speed, address a different problem with different evaluation criteria. A direct comparison would be inappropriate and would obscure the core contribution of our work. We therefore maintain our focus on comparisons with other state-of-the-art exact methods.

---

> ### Author Response · Authors · 2025-12-02
> **Weakness 2**
>
> The relationship between $p$ and $k$ is closely linked to the distribution of centers, but through an intermediary (the Elkan algorithm), we can gain a deeper understanding of $p$.
> Let $q$ be the average number of candidate centers each sample needs to consider in the Elkan algorithm. From Figure 1, it is not difficult to see that the volume $V_a$ considered by AKM and the volume $V_e$ considered by Elkan satisfy the following relationship:
>
> $$
> \frac{1}{2^d} V_e \leq V_a \leq \frac{2}{2^d} V_e.
> $$
>
> Assuming the centers are uniformly distributed, we obtain:
>
> $$
> \frac{1}{2^d} q \leq p \leq \frac{2}{2^d} q.
> $$
>
> From this, it can be observed that our filtering effectiveness shows significant improvement compared to the Elkan algorithm.

---

### Author Response · Authors · 2025-12-02
**Summary of Responses**

We have carefully revised our manuscript and prepared detailed responses to all comments raised by the reviewers. The revised manuscript has been uploaded to the submission system. Below is a summary of the key points addressed:

**For Reviewer qQbA (Rating: 2):** This reviewer raised concerns about the derivation of Equation (10) and the time complexity analysis.
- We have addressed and resolved the reviewer's questions regarding the time complexity calculation.
- For the derivation of Equation (10), we have provided a detailed, step-by-step explanation in our response.

**For point 1, the reviewer replied: “Thanks for your comments. I am satisfied with the running time now.” For point 2, the reviewer provided no further response.**

---
**For Reviewer NsYa (Rating 4):** We believe the reviewer may not possess adequate familiarity with this research domain.
In our response, we have explained point by point why his/her comments are unhelpful to our submission.
- TILB K-means is not a clustering algorithm, and therefore cannot be included as a comparative method.
- Our core filtering formula required over a page of rigorous proof, which the reviewer dismissively called a "minor modification." This characterization is entirely unacceptable.
- Our algorithm maintains strong acceleration performance even in high-dimensional settings, and corresponding experimental results have been added to the Comment.
- Criticizing our Euclidean-based method for not extending to other distance metrics would logically disqualify nearly all prior work in this well-established subfield.

**To date, the reviewer has not submitted any follow-up comments**

---
**For Reviewer 5Q16 (Rating 6):**
- We did not include approximate K-means algorithms in our baseline because we focus only on studies that aim to accelerate exact K-means.
- We provided a detailed theoretical analysis to elaborate on the acceleration mechanism and its expected performance gains.
- We validated the algorithm’s performance through experiments, especially when k is large.

**After we addressed point 3, the reviewer replied, “Thanks for the clarifications. I prefer to maintain my initial evaluations at this stage.” The reviewer was most concerned about point 2, but after our response, He/She were unable to add further comments.**

---
**For Reviewer  joh2(Rating 8):**
- We have clarified the details of how the methods compare in time and space complexity, and in empirical speed.
- We have revised the formatting of the manuscript.
- We have provided a clearer explanation of the purpose of Figure 1(b).
- Following the reviewer's suggestion, we have added the Elkan algorithm as a baseline method in our comparative analysis.
- We have updated Tables 3-6 as suggested, and the metrics are now presented in percentage terms relative to Lloyd’s algorithm as the baseline
- We have responded point-by-point to all other specific questions and suggestions raised by the reviewer.

**To date, the reviewer has not submitted any follow-up comments**

---

### Meta-Review · Area_Chair_Vdfh · 2025-12-28

**Summary:**

We received mixed reviews for this paper, and most are negative. The concerns raised by the negative ones are valid. The authors seem to do a decent job in the rebuttal, and I find it convincing. I would nonetheless suggest to accept the paper.

**Reviewer Concerns:**

There are multiple concerns suggested by the reviewers. I find the status mentioned in the authors' post https://openreview.net/forum?id=BdhQWT0y9s&noteId=8oyoXSiPmU a great summarization.

Overall, I think the concern about whether the method works well for large k and/or d is still an outstanding issue and is only partially addressed -- the authors addressed these by showing promising experiment results, but no theoretical discussion is provided. In fact, an intuitive explanation is useful, especially for the high-dimensional case.

**Reviewer Scores:**

Reviewer 5Q16 with score 6, and Reviewer joh2 with score 8, will not change the score (and Reviewer 5Q16 already explicitly mentioned this). Reviewer NsYa (with score 4) and Reviewer qQbA (with score 2) may raise their score, because solid new evidence has been provided to clarify their concerns.

---

### Decision · Program_Chairs · 2026-01-26

Accept (Poster)